# Pathogenic ACVR1[R206H] activation by Activin A-induced receptor clustering and autophosphorylation

Anassuya Ramachandran[1,*,†,‡] [iD], Merima Mehić[1,†] [iD], Laabiah Wasim[2], Dessislava Malinova[2] [iD], Ilaria Gori[1] [iD], Beata K Blaszczyk[3], Diana M Carvalho[4], Eileen M Shore[5] [iD], Chris Jones[4] [iD], Marko Hyvönen[3] [iD], Pavel Tolar[2,#] [iD] & Caroline S Hill[1,**] [iD]

## Abstract

Fibrodysplasia ossificans progressiva (FOP) and diffuse intrinsic pontine glioma (DIPG) are debilitating diseases that share causal mutations in ACVR1, a TGF-β family type I receptor. ACVR1[R206H] is a frequent mutation in both diseases. Pathogenic signaling via the SMAD1/5 pathway is mediated by Activin A, but how the mutation triggers aberrant signaling is not known. We show that ACVR1 is essential for Activin A-mediated SMAD1/5 phosphorylation and is activated by two distinct mechanisms. Wild-type ACVR1 is activated by the Activin type I receptors, ACVR1B/C. In contrast, ACVR1[R206H] activation does not require upstream kinases, but is predominantly activated via Activin A-dependent receptor clustering, which induces its auto-activation. We use optogenetics and live-imaging approaches to demonstrate Activin A-induced receptor clustering and show it requires the type II receptors ACVR2A/B. Our data provide molecular mechanistic insight into the pathogenesis of FOP and DIPG by linking the causal activating genetic mutation to disrupted signaling.

**Keywords** Activin A; ACVR1[R206H]; DIPG; FOP; receptor clustering

**Subject Categories** Cancer; Signal Transduction

The EMBO Journal (2021) 40: e106317

## Introduction

Fibrodysplasia ossificans progressiva (FOP) and diffuse intrinsic pontine glioma (DIPG) are devastating human diseases, both associated with activating mutations in ACVR1, a type I TGF-β family receptor (Taylor *et al*, 2014b). FOP is a monogenic, autosomal dominant disease caused by germline mutations in ACVR1 and characterized by progressive, episodic heterotopic ossification of soft tissue (Shore *et al*, 2006; Furuya *et al*, 2008; Kaplan *et al*, 2009). DIPG, in contrast, is an invariably fatal childhood glioma arising in the ventral pons of the hindbrain, with a median survival of 9–12 months. Approximately 20% of DIPG tumors harbor somatic ACVR1 mutations (Buczkowicz *et al*, 2014; Fontebasso *et al*, 2014; Taylor *et al*, 2014a; Wu *et al*, 2014; Mackay *et al*, 2017).

TGF-β family ligands require two different receptors for signaling, type I and type II receptors, which are thought to function as a heterotetramer, comprising two type I and two type II receptors. Ligand binding brings the receptors together, whereupon the constitutively active type II receptor phosphorylates and activates the type I receptor (Wrana *et al*, 1994). The activated type I receptors then phosphorylate the intracellular effectors of the pathway, the receptor-activated SMADs (R-SMADs). Phosphorylated R-SMADs form complexes with SMAD4 and SMAD heterotrimers accumulate in the nucleus to regulate gene expression (Miller & Hill, 2016). There are five type II receptors and seven type I receptors, and different ligands use different combinations of receptors to signal. The identity of the SMADs phosphorylated by a particular ligand–receptor complex is dictated by the type I receptor engaged (Massague, 2012). Thus, ACVR1B and ACVR1C, which bind Activins and Nodal, and TGFBR1, which binds TGF-βs, induce phosphorylation of SMAD2 and SMAD3. The other type I receptors, which bind BMPs and GDFs, induce phosphorylation of SMAD1, SMAD5, and SMAD9 (Massague, 2012). ACVR1 is in this latter group and has a preference for BMPs (Miller & Hill, 2016).

An exception to these rules emerged with the discovery that TGF-β signaling through ACVRL1 and ACVR1 led to SMAD1/5

1   Developmental Signalling Laboratory, The Francis Crick Institute, London, UK
2   Immune Receptor Activation Laboratory, The Francis Crick Institute, London, UK
3   Department of Biochemistry, University of Cambridge, Cambridge, UK
4   Division of Molecular Pathology, The Institute of Cancer Research, Sutton, UK
5   Departments of Orthopaedic Surgery and Genetics, Perelman School of Medicine, University of Pennsylvania, Philadelphia, PA, USA
    *Corresponding author. Tel: +64 9 923 1431; E-mail: anassuya.ramachandran@auckland.ac.nz
    **Corresponding author. Tel: +44 203 796 1251; E-mail: caroline.hill@crick.ac.uk
    †These authors contributed equally to this work
    ‡Present address: Department of Molecular Medicine and Pathology, University of Auckland, Auckland, New Zealand
    #Present address: Division of Infection and Immunity, Institute of Immunity and Transplantation, University College, London, UK

phosphorylation (Goumans *et al*, 2002; Lebrin *et al*, 2004; Daly *et al*, 2008; Ramachandran *et al*, 2018). TGF-β-induced SMAD1/5 phosphorylation is required for cell proliferation in endothelial cells (Goumans *et al*, 2002; Lebrin *et al*, 2004) and for anchorage-independent growth, cell migration, and epithelial-to-mesenchymal transition (EMT) in other cell types (Miettinen *et al*, 1994; Daly *et al*, 2008; Liu *et al*, 2009; Ramachandran *et al*, 2018). The underlying mechanism of receptor activation in this case represents a departure from the classical mechanism and involves a type I receptor activating another type I receptor. We recently demonstrated that TGFBR1, activated as a result of TGF-β stimulation, can phosphorylate and activate ACVR1, which subsequently phosphorylates SMAD1/5 (Ramachandran *et al*, 2018). It is not yet known whether this mechanism of activating ACVR1 is unique to TGF-β or whether other receptor complexes that normally signal through SMAD2/3 can also fulfill this function.

The mutations in ACVR1 that lead to FOP occur in the cytoplasmic portion of the receptor and are concentrated around both the GS domain, a glycine-serine rich juxtamembrane region, and the kinase domain (Han *et al*, 2018). Although approximately 13 mutations have been identified, almost 97% of cases of FOP have a single amino acid change, Arg206His (hereafter ACVR1$^{R206H}$) (Shore *et al*, 2006; Han *et al*, 2018). This same mutation accounts for about a fifth of DIPG mutations (Han *et al*, 2018).

ACVR1 mutations in FOP and DIPG, and in particular ACVR1$^{R206H}$, lead to enhanced SMAD1/5-induced signaling (Fukuda *et al*, 2009; Shen *et al*, 2009; van Dinther *et al*, 2010; Mucha *et al*, 2018). However, the underlying mechanism has proven controversial. One possible explanation has centered around the immunophilin FKBP1A (also known as FKBP12), which binds the GS domain of the type I receptors, maintaining them in an inactive conformation (Wang *et al*, 1994; Okadome *et al*, 1996). Disease mutations in ACVR1 were thought to disrupt binding of FKBP1A, thereby removing pathway inhibition. However, the vast majority of ACVR1 mutants retain the ability to bind FKBP1A and can be inhibited by it, suggesting that this mechanism does not explain the enhanced activity of ACVR1$^{R206H}$ (Groppe *et al*, 2011; Chaikuad *et al*, 2012; Machiya *et al*, 2018).

The roles of type II receptors and activating ligands in the activity of ACVR1$^{R206H}$ are also not fully understood. In *Drosophila*, the hyperactivity of transgenic ACVR1$^{R206H}$ is dependent on the presence of a type II receptor (Le & Wharton, 2012). Similarly, transgenic mice overexpressing ACVR1$^{Q207D}$, another activating mutation, only developed heterotopic ossification in the presence of the type II receptors ACVR2A and BMPR2, demonstrating a requirement of the type II receptors for the pathology of mutant ACVR1 in FOP (Bagarova *et al*, 2013). Whether the type II receptors perform a catalytic or structural role has remained unclear. As ACVR1 is best characterized as a BMP receptor (Luo *et al*, 2010), initial work focused on the BMPs as the activating ligands (Fukuda *et al*, 2009; Shen *et al*, 2009; Song *et al*, 2010; van Dinther *et al*, 2010; Chaikuad *et al*, 2012). However, two recent studies using murine models and patient-derived induced pluripotent stem cells concluded that Activin A was the relevant ligand signaling through ACVR1$^{R206H}$ in FOP (Hatsell *et al*, 2015; Hino *et al*, 2015). In both cases, Activin A exhibited strong differential signaling through the mutant receptor, compared to the wild-type receptor, leading to robust SMAD1/5 phosphorylation, induction of downstream target genes, and the

initiation of heterotopic ossification in mice. Notably, this pathological signaling could be blocked using the soluble type II receptor extracellular domains, acting as ligand traps or Activin A neutralizing antibodies (Hatsell *et al*, 2015; Lees-Shepard *et al*, 2018).

Here, we unravel the mechanism whereby Activin A signals through ACVR1$^{R206H}$. We use extensive genome editing in HEK293T cells, as well as glioma cells from DIPG patients to both model the disease mutation, ACVR1$^{R206H}$, and resolve the role of additional receptors in Activin A-induced ACVR1$^{R206H}$ activity. We provide strong evidence that type I receptor activation of other type I receptors, as we previously established for TGF-β, is a general mechanism and not restricted to TGF-β receptors. We demonstrate that wild-type ACVR1 (hereafter ACVR1$^{WT}$) is essential for Activin A-induced SMAD1/5 phosphorylation, and that similar to TGFBR1, ACVR1B/C activate ACVR1$^{WT}$ downstream of Activin A to mediate this phosphorylation. In contrast, ACVR1$^{R206H}$ is predominantly auto-activated by clustering, independent of the kinase activity of type II receptors or ACVR1B/C. Finally, we use live-imaging approaches to demonstrate Activin A-induced receptor clustering and show that this requires the type II receptors, ACVR2A/B. These results elucidate the signaling mechanism of ACVR1$^{R206H}$, which is crucial for understanding the pathogenesis of FOP and DIPG.

# Results

## ACVR1$^{R206H}$ exhibits increased responsiveness to Activin A, relative to wild-type ACVR1

To dissect the mechanism whereby the mutant ACVR1$^{R206H}$ receptor signals, we introduced this point mutation into the endogenous ACVR1 locus in a clone of HEK293T cells (hereafter referred to as parental) using CRISPR/Cas9 genome editing. A heterozygous clone (HET) and several homozygous clones (HOM1 and HOM2) were obtained (Fig EV1A and B). We first characterized the responsiveness of these clones to different TGF-β family ligands (Activin A, Activin B, BMP4, BMP7, BMP4/7 heterodimer, and BMP2), using SMAD1/5 phosphorylation (pSMAD1/5) as a readout. Treatment of parental cells for 1 h with all these ligands resulted in induction of pSMAD1/5, with Activin A being the least efficient (Figs 1A and C, and EV1C). Strikingly, cells that contained the ACVR1$^{R206H}$ allele (HET, HOM1, and HOM2) all showed substantially increased sensitivity to Activin A (Figs 1A and EV1C), whereas induction of pSMAD1/5 in response to the other ligands in HET, HOM1, and HOM2 cells was comparable to that in the parental cells (Figs 1A and C, and EV1C). We also generated several ACVR1 knockout clones (KO1 and KO2) (Appendix Supplementary Methods). These knockout cells were markedly inhibited in their response to Activin A, Activin B, BMP4/7, and to a lesser extent BMP7, but not to BMP4 and BMP2 (Figs 1A and C, and EV1C). As expected, Activin A and Activin B also led to SMAD2 phosphorylation, that was unaffected by the presence of AVCR1$^{R206H}$ or loss of ACVR1 (Figs 1A and EV1C). Thus, ACVR1$^{WT}$ is essential for the induction of pSMAD1/5 in response to Activin A, Activin B, and BMP4/7, and ACVR1$^{R206H}$ selectively confers increased responsiveness to Activin A.

To understand how ligand dose affected signaling, we compared the responses of parental, HET, and HOM1 cells to increasing ligand doses, focusing on Activin A and BMP4/7 (as a control ligand that

also signals through ACVR1). In parental cells, a low level of pSMAD1/5 was achieved with 1.25 ng/ml Activin A and this was essentially unchanged at higher ligand concentrations (Fig 1B). However, in both HET and HOM1 cells, Activin A led to a dose-dependent quantitative increase in pSMAD1/5 that plateaued at

around 5 ng/ml Activin A (Fig 1B). Activin A led to similar SMAD2 phosphorylation in parental, HET, and HOM1 cells, plateauing in the range of 0.625–1.25 ng/ml (Fig 1B). Parental, HET, and HOM1 cells all displayed a similar dose response to BMP4/7, with SMAD1/5 phosphorylation plateauing at around 2.5–5 ng/ml BMP4/7 (Fig 1

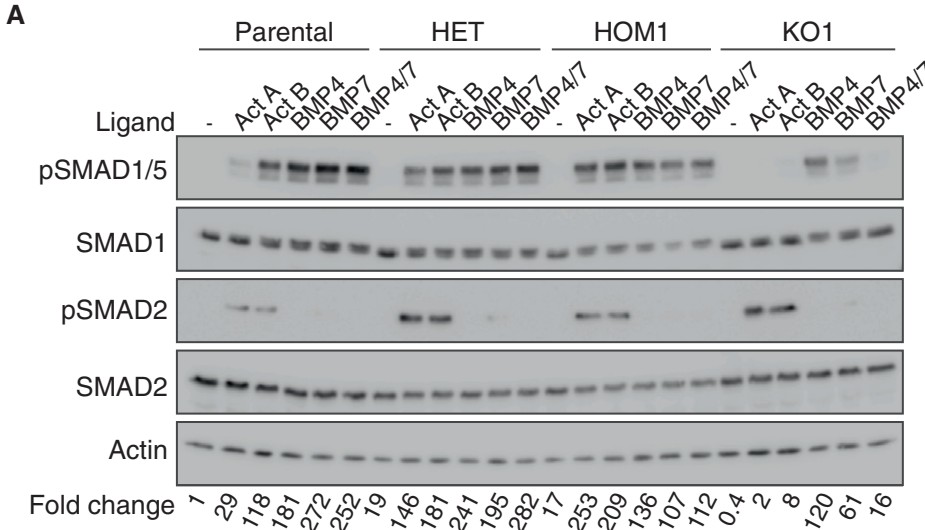

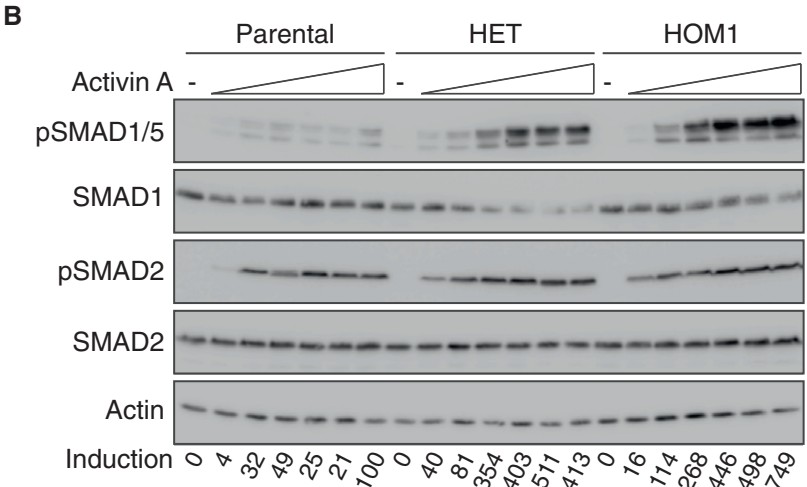

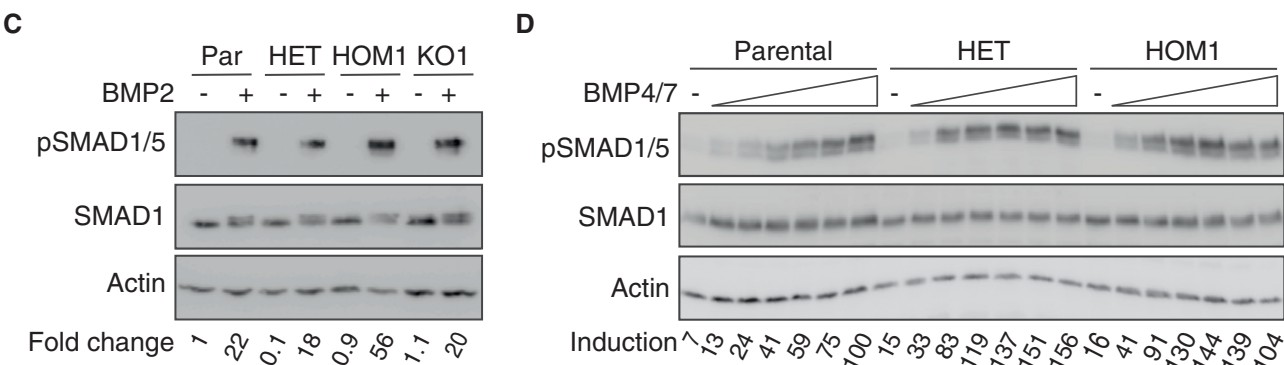

**Figure 1.**

**Figure 1.   Characterization of ACVR1 responsiveness to TGF-β family ligands.**

A   Parental HEK293T, HET, HOM1, and KO1 cells were stimulated with the indicated ligands for 1 h.
B   Cells were stimulated with increasing (doubling) doses of Activin A (0.625–20 ng/ml) for 1 h.
C   Cells were stimulated with BMP2 for 1 h.
D   Cells were stimulated with increasing (doubling) doses of BMP4/7 (0.625–20 ng/ml) for 1 h.

Data information: In all cases, whole cell extracts were Western-blotted using the indicated antibodies. Representative experiments are shown. The levels of pSMAD1/5 relative to actin were determined and are expressed as a fold change relative to the parental HEK293T untreated sample in (A and C), or normalized to the 20 ng/ml ligand dose in parental HEK293T cells in (B and D). Act, Activin A; Par, parental HEK293T cells; and KO1, ACVR1 KO clone 1.
Source data are available online for this figure.

D), indicating that signaling via this ligand is not sensitive to the mutation status of ACVR1. Thus, in parental cells, the similarity of pSMAD1/5 and pSMAD2 induction with increasing Activin A concentrations suggested the involvement of shared activated receptors. In contrast, the graded phosphorylation of SMAD1/5 with increasing Activin A dose in cells containing ACVR1$^{R206H}$, not seen with pSMAD2, suggested that Activin A triggers an alternative receptor activation mechanism in cells bearing this mutation.

### ACVR1$^{R206H}$ confers an extended duration of SMAD1/5 phosphorylation relative to ACVR1$^{WT}$

To investigate the effect of the R206H mutation in ACVR1 on signaling dynamics, we performed time courses of Activin A and Activin B stimulation in the different cell lines, with BMP2 treatment time courses as a control. Parental cells exhibited a transient pSMAD1/5 response to Activin A and B, whereas in HET, HOM1, and HOM2 cells, the duration of SMAD1/5 phosphorylation was extended. This was most prominent in the HOM1 cells, whereas HET cells displayed kinetics intermediate between HOM1 and parental cells (Figs 2A and EV2A). These phosphorylation kinetics are also mirrored in the expression of pSMAD1/5 target genes, *ID1*, *ID3*, and *ATOH8* (Grönroos *et al*, 2012; Ramachandran *et al*, 2018), where the induction of target gene expression seen in parental cells is more robust and sustained in HET and HOM1 cells (Fig EV2B). The differential pSMAD1/5 observed between the parental and HOM1 cells was temporary; however, as by 24 h of Activin A treatment, SMAD1/5 phosphorylation in HOM1 cells had returned to basal levels seen in parental cells (Fig EV2C). As expected, the presence of the point mutation in ACVR1 did not alter responsiveness to BMP2, which signals through other TGF-β family type I receptors, over the time course (Fig 2B).

To shed light on the mechanism governing pSMAD1/5 signaling duration, we investigated whether the SMAD2/3 arm of signaling was involved. Strikingly, combined knockout of SMAD2 and SMAD3 in parental cells extended the duration of Activin A-mediated SMAD1/5 phosphorylation (Figs 2C and EV2D, Appendix Supplementary Methods), suggesting that downstream targets of this signaling arm could be responsible. The most likely targets were the I-SMADs SMAD6 and SMAD7, as these are induced in response to Activin A in parental cells (Fig EV2E), and known to be involved in mediating receptor turnover (reviewed in Miller and Hill (2016)). We therefore knocked out SMAD6 and SMAD7, individually and simultaneously using CRISPR/Cas9 genome editing (Appendix Supplementary Methods). Combined knockout of both SMAD6 and SMAD7 also extended the duration of Activin A-induced pSMAD1/5 (Fig 2C), while knocking out each alone had little effect (Fig EV2F). Thus, in parental cells, SMAD2/3 signaling, at least in part via the induction of SMAD6/7, acts to specifically attenuate SMAD1/5 phosphorylation downstream of Activin A.

We next addressed why ACVR1$^{R206H}$ mediated an extended signaling duration compared to ACVR1$^{WT}$. While TGF-β family signaling is initiated at the cell surface, the majority of signaling occurs in the internalized endosomal compartment (Vizán *et al*, 2013; Miller *et al*, 2018; Miller *et al*, 2019). We therefore investigated whether the longer duration of ACVR1$^{R206H}$ signaling might result from extended signaling from these intracellular compartments. Cells were pulsed with Activin A for 1 h and then chased with its specific antagonist follistatin (Nakamura *et al*, 1990) for different durations, up to 4 h. As the addition of follistatin prevents further binding of Activin A to receptors at the cell surface, any SMAD phosphorylation detected in these conditions would be the result of ongoing signaling from endosomes. Parental, HET, and HOM1 cells all demonstrated similar half-lives for phosphorylated SMAD2 and SMAD1/5 (Fig EV3A). Thus, ACVR1$^{R206H}$ does not obviously alter signaling dynamics from endosomes. This left open the possibility that the underlying mechanism of altered signaling dynamics could be at the level of surface receptors.

**Figure 2.   ACVR1$^{R206H}$ and SMAD2/3 signaling influence the kinetics of SMAD1/5 phosphorylation.**

A, B   Parental HEK293T, HET, and HOM1 cells were treated with Activin A or Activin B (A) or BMP2 (B) for the indicated times. Quantifications are shown on the right.
C      Parental HEK293T, S2/3 dKO1, and S6/7 dKO1, and S6/7 dKO2 cells were treated with Activin A for the indicated times.
D      Parental HEK293T, S2/3 dKO1, and HOM1 cells were treated with Activin A for the indicated times.

Data information: In (A–C), quantifications are the average normalized intensities ± SEM of between 3 and 10 (A and C) or 2 (B) independent experiments, and in the case of Activin A and Activin B, include a second ACVR1$^{R206H}$ homozygous clone (HOM2, see Fig EV2A). In (A), quantifications are the levels of pSMAD1/5 normalized to actin, and in (B and C), the quantifications are expressed as fold changes relative to 1 h of ligand stimulation. ns, not significant; *$P < 0.05$; **$P < 0.01$; ***$P < 0.001$; and ****$P < 0.0001$. The *P*-values are from two-way ANOVA with Dunnett's *post hoc* test. In (D), quantifications are from a representative experiment and are the intensities of pSMAD1/5 normalized to actin, expressed as fold change relative to untreated cells for each clone. In all panels, Western blots of whole cell lysates were probed with the antibodies indicated. S2/3 dKO1, SMAD2/3 double knockout clone 1 and S6/7 dKO1/2, SMAD6/7 double knockout clones 1 and 2.
Source data are available online for this figure.

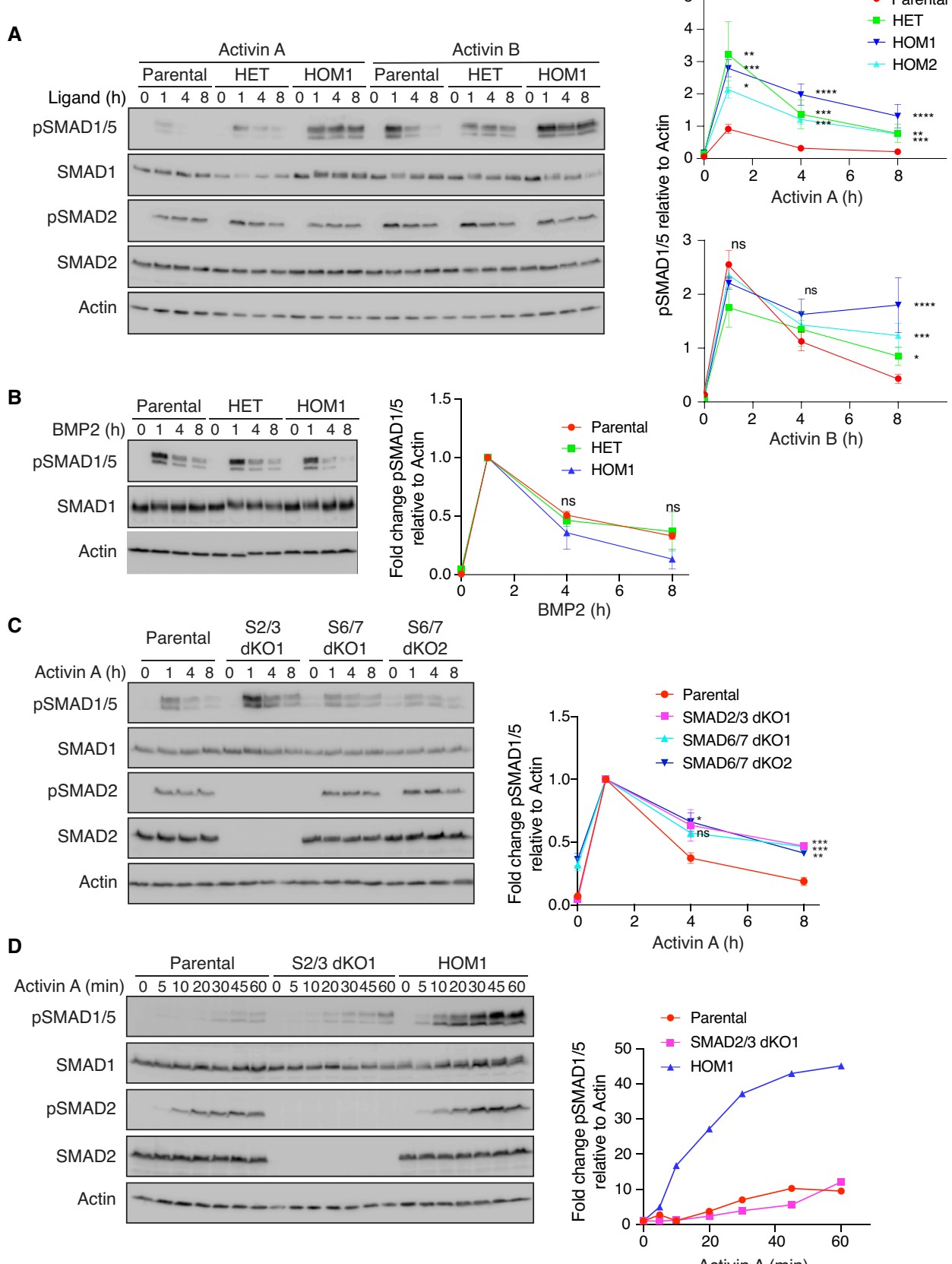

**Figure 2.**

To address this, we investigated whether ACVR1 was more rapidly depleted from the cell surface after Activin A treatment in parental cells, compared with HOM1 cells. To this end, we treated parental and HOM1 cells with Activin A for 2 h and then added the antagonist follistatin for a further 2 h to reduce phosphorylated R-SMAD levels back to basal levels (Fig EV3B). The follistatin was then washed out and the cells re-stimulated with Activin A for 1 h. In both parental and HOM1 cells, Activin A led to re-phosphorylation of both SMAD1/5 and SMAD2, demonstrating that ACVR1$^{WT}$ and ACVR1$^{R206H}$ (and the classic Activin type I receptors ACVR1B and ACVR1C) are still present at the cell surface following an acute Activin A treatment (Fig EV3B). Moreover, we showed that ACVR1$^{WT}$ and ACVR1$^{R206H}$ at the cell surface even after 8-h Activin treatment can be engaged by other ligands, including BMP4/7, to induce SMAD1/5 phosphorylation (Fig EV3C). Thus, the intracellular signaling dynamics and recycling of receptors to the cell surface are broadly similar for ACVR1$^{WT}$ and ACVR1$^{R206H}$.

## ACVR1$^{R206H}$ alters the catalytic efficiency of ACVR1 in response to Activin A

We have shown that the R206H mutation in ACVR1 increases both the magnitude and the duration of SMAD1/5 phosphorylation in response to Activin A. To begin to determine the underlying mechanism, we performed a high-resolution time course of Activin A induction in parental, HET, and HOM1 cells, sampling every 5–15 min for the first hour of signaling. A higher initial rate of SMAD1/5 phosphorylation was measured in the HET and HOM1 cells, relative to parental cells, suggesting that the R206H mutation endowed an increase in the catalytic efficiency of ACVR1 (Fig 2D, Appendix Fig S1A). As a control, we found that the initial rates of BMP2-mediated pSMAD1/5 were unaffected by the presence of the mutation in ACVR1 (Appendix Fig S1B). Furthermore, the extended pSMAD1/5 duration in response to Activin A in the SMAD2/3 dKO cells was not due to an increased catalytic activity of ACVR1$^{WT}$, as the time course of Activin A-induced pSMAD1/5 in these cells was similar to that seen in parental cells (Fig 2D). Thus, although both ACVR1$^{R206H}$ and loss of SMAD2/3 lead to an increase in levels and duration of Activin A-induced pSMAD1/5, the underlying mechanisms are distinct.

## ACVR1B/1C are required for the activation of ACVR1$^{WT}$, but not ACVR1$^{R206H}$

We next investigated how Activin A induces signaling through ACVR1$^{WT}$ and ACVR1$^{R206H}$. We hypothesized that Activin A might induce the activation of ACVR1 by a mechanism analogous to the one we previously described for TGF-β (Ramachandran et al, 2018). In that case, TGF-β binding to TGFBR2 leads to the canonical activation of TGFBR1, which subsequently phosphorylates and activates ACVR1. We thus investigated whether the Activin type I receptors, ACVR1B and ACVR1C, were involved in Activin A-induced signaling via wild-type and mutant ACVR1. Additionally, we noted above that the wild-type and mutant ACVR1 behaved differently to escalating doses of Activin A (Fig 1B), with ACVR1$^{WT}$ being relatively insensitive to the dose of Activin A, while the mutant receptor responded in a dose-dependent manner. This suggested that higher doses of Activin A might activate the mutant receptor through a distinct mechanism to low doses of Activin A.

To test these ideas, we treated parental, HET, and HOM1 cells with either a high (20 ng/ml) or a low (1.25 ng/ml) dose of Activin A, in the absence or presence of SB-431542, that inhibits the kinase activity of ACVR1B/C (Inman et al, 2002), or LDN-193189, that inhibits the kinase activity of ACVR1/BMPR1A/BMPR1B (Cuny et al, 2008). In all cells and at both Activin A doses, SB-431542 completely prevented Activin A-induced pSMAD2, demonstrating that it is sufficient to block all ACVR1B/C activity. In parental cells, SB-431542 also completely inhibited the phosphorylation of SMAD1/5 at both concentrations of Activin A (Fig 3A and B). Similarly, in HET and HOM1 cells, pSMAD1/5 levels induced with a low dose of Activin A were substantially reduced by treatment with SB-431542 (Fig 3A). However, strikingly, in HET, and in particular in HOM1 cells treated with a high dose of Activin A, significant pSMAD1/5 persisted, even after treatment with SB-431542 (Fig 3A and B). In all cell lines and at both Activin A doses, treatment with LDN-193189 completely blocked SMAD1/5 phosphorylation downstream of Activin A, without affecting SMAD2 phosphorylation. Similarly, LDN-193189, but not SB-431542, completely blocked BMP4/7-induced pSMAD1/5 (Fig 3B). Thus, we conclude that in all cases, Activin A- and BMP4/7-induced pSMAD1/5 requires the kinase activity of ACVR1. For ACVR1$^{WT}$, Activin A-induced pSMAD1/5 additionally requires the kinase activity of ACVR1B/C. For the mutant receptor, a similar mechanism operates at low doses of Activin A, but at higher Activin A concentrations, activation of ACVR1$^{R206H}$ occurs via a second mechanism that is largely independent of ACVR1B/C.

To investigate whether these mechanisms are relevant in DIPG, we treated cell lines derived from DIPG patients that are either ACVR1$^{+/+}$ (ICR-B169) or ACVR1$^{+/R206H}$ (HSJD-DIPG-007) (Carvalho et al, 2019; C. Jones, personal communication) with Activin A. ICR-B169 cells did not detectably induce pSMAD1/5 in response to Activin A (Fig 3C). However, in agreement with observations in HET and HOM1 cells, Activin A induced pSMAD1/5 in HSJD-DIPG-007 cells that was inhibited by LDN-193189, but unaffected by SB-431542, although the latter completely inhibited Activin A-induced pSMAD2 (Fig 3C). In addition, BMP4/7-induced pSMAD1/5 was inhibited by LDN-193189 in both ICR-B169 and HSJD-DIPG-007 cells (Fig 3C). Unexpectedly, BMP4/7 also induced low levels of pSMAD2 in HSJD-DIPG-007 cells. We have previously observed that recombinant ACVR1 is capable of efficiently phosphorylating SMAD2 in vitro (Ramachandran et al, 2018) raising the possibility that ACVR1$^{R206H}$ may confer a neofunction to BMP4/7.

To further confirm these results obtained with small molecule inhibitors, we took a genetic approach and simultaneously knocked out ACVR1B and ACVR1C in both parental and HOM1 cells using CRISPR/Cas9 genome editing (Appendix Supplementary Methods). As expected, in parental cells lacking both ACVR1B/C, Activin A-induced pSMAD1/5 and pSMAD2 were lost, while SMAD1/5 phosphorylation in response to BMP2 was unaffected (Fig 3D). Strikingly, in HOM1 cells, knockout of both ACVR1B/C led to a complete loss of pSMAD2 but did not inhibit pSMAD1/5 phosphorylation in response to Activin A (Fig 3D). Again, BMP2-mediated SMAD1/5 phosphorylation was unaffected by the absence of ACVR1B/C.

The data described above demonstrate that higher doses of Activin A induce pSMAD1/5 via two different mechanisms, depending on the mutation status of ACVR1; ACVR1$^{WT}$ signaling requires ACVR1B/C, while ACVR1$^{R206H}$ does not. We therefore considered

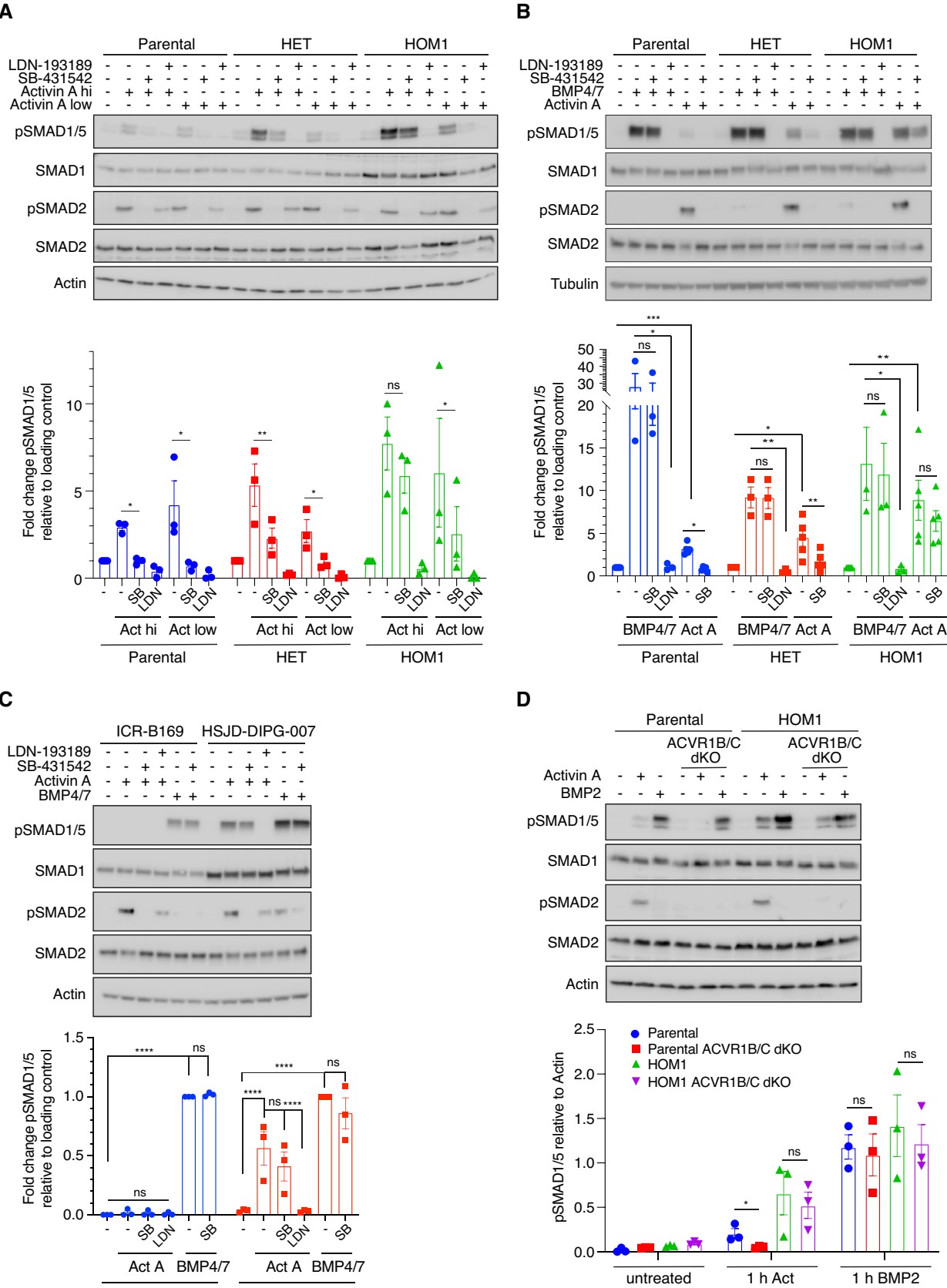

**Figure 3.**

**Figure 3.  Distinct type I receptor requirements for Activin A-mediated SMAD1/5 phosphorylation by ACVR1$^{WT}$ and ACVR1$^{R206H}$.**

A   Parental HEK293T, HET, and HOM1 cells were treated with high (20 ng/ml) or low (1.25 ng/ml) concentrations of Activin A for 1 h, in the presence or absence of either 1 μM LDN-193189 or 10 μM SB-431542. Quantifications are the levels of pSMAD1/5 normalized to the loading control (actin), expressed as fold change relative to untreated for each clone. Quantifications are the average normalized intensities ± SEM of 3 independent experiments.

B   Parental HEK293T, HET, and HOM1 cells were treated with either Activin A or BMP4/7 (both at 20 ng/ml) for 1 h, in the presence or absence of either 1 μM LDN-193189 or 10 μM SB-431542. Quantifications are the levels of pSMAD1/5 normalized to the loading control (actin or tubulin), expressed as fold change relative to untreated for each clone. Quantifications are the average normalized intensities ± SEM of 3–5 independent experiments. Note that the Activin A quantifications also includes data generated with high dose Activin A in Fig 3A as they are independent replicates.

C   ICR-B169 and HSJD-DIPG-007 cells were treated with either Activin A or BMP4/7 (both at 20 ng/ml) for 1 h, in the presence or absence of either 1 μM LDN-193189 or 10 μM SB-431542. Quantifications are the levels of pSMAD1/5 normalized to the loading control (actin) expressed as fold change relative to BMP4/7 induction in each clone and are the average normalized intensities ± SEM of 3 independent experiments.

D   Parental HEK293T, ACVR1B/C dKO HOM1, and HOM1 ACVR1B/C dKO cells were treated with 20 ng/ml Activin or BMP2 for 1 h. Quantifications are the average normalized intensities of pSMAD1/5 normalized to actin ± SEM of 3 independent experiments.

Data information: In all panels, Western blots of whole cell lysates were probed with the antibodies indicated. hi, high; Act, Activin A; SB, SB-431542; and LDN, LDN-193189. The *P*-values (for A, B, and D) are from a *t*-test with Holm–Sidak correction, for comparison between indicated pairs. ns, not significant; *P < 0.05; **P < 0.01; ***P < 0.001; and ****P < 0.0001. The *P*-values in (C) are from two-way ANOVA with Tukey's *post hoc* correction.

Source data are available online for this figure.

the possibility that ACVR1$^{R206H}$ could similarly be activated by TGF-β via a mechanism independent of TGFBR1. HEK293T cells respond very poorly to TGF-β stimulation (Levy *et al*, 2007), so we used mouse embryonic fibroblasts (MEFs) from ACVR1$^{+/R206H}$ mice and MEFs derived from their matched wild-type littermates (ACVR1$^{+/+}$) (Culbert *et al*, 2014) for this experiment. TGF-β induced pSMAD1/5 and pSMAD2 in both lines (Appendix Fig S2). In ACVR1$^{+/+}$ MEFs, treatment with the TGFBR1 kinase inhibitor, SB-431542, inhibited both pSMAD1/5 and pSMAD2 downstream of TGF-β to basal levels, as observed previously for other cell lines (Ramachandran *et al*, 2018). Strikingly, in ACVR1$^{+/R206H}$ MEFs, SMAD1/5 phosphorylation downstream of TGF-β was unaffected by SB-431542, but only blocked by LDN-193189 (Appendix Fig S2). Thus, multiple noncanonical ligands can trigger the additional mechanism of receptor activation conferred by ACVR1$^{R206H}$.

## The kinase activity of type II receptors is not required for ACVR1$^{R206H}$ function

To address the role of the type II receptors in Activin A-induced signaling via ACVR1$^{R206H}$, we deleted ACVR2A and/or ACVR2B in both parental HEK293T and HOM1 cells using CRISPR/Cas9 (Appendix Supplementary Methods). In both cell lines, ACVR2A contributed more to Activin A-induced SMAD2 and SMAD1/5 phosphorylation than ACVR2B. Combined loss of ACVR2A/B led to a loss of pSMAD2 and pSMAD1/5 in both lines (Fig 4A). Importantly, phosphorylation of SMAD2 was rescued in ACVR2A/B dKO lines (both parental and HOM1) by transfecting in wild-type ACVR2A (ACVR2A$^{WT}$) (Fig 4B and C).

To disentangle the ligand binding function of ACVR2A from its catalytic activity, we used a kinase-dead ACVR2A, in which the invariant lysine in the ATP binding pocket had been mutated to arginine (ACVR2A$^{K219R}$, hereafter called ACVR2A$^{KR}$) (Bagarova *et al*, 2013). In ACVR2A/B dKO parental cells, only ACVR2A$^{WT}$, but not ACVR2A$^{KR}$, could rescue both pSMAD2 and pSMAD1/5 (Fig 4B). Similarly, in ACVR2A/B dKO HOM1 cells, ACVR2A$^{WT}$ could restore both pSMAD2 and pSMAD1/5. In marked contrast, in ACVR2A/B dKO HOM1 cells, ACVR2A$^{KR}$ restored SMAD1/5 phosphorylation, but not SMAD2 phosphorylation (Fig 4C). Thus, in parental cells, phosphorylation of SMAD2 and SMAD1/5 are linked and share aspects of a common receptor activation mechanism for signaling. In contrast, in ACVR1$^{R206H}$-containing cells, SMAD2 and SMAD1/5 phosphorylation are uncoupled from each other, and SMAD1/5 phosphorylation is achieved by a mechanism distinct from the classic type II–type I receptor activation cascade.

## Clustering of ACVR1$^{R206H}$ is sufficient to initiate signaling

The ability of ACVR2A$^{KR}$ to restore pSMAD1/5 in Activin A-induced ACVR2A/B dKO HOM1 cells and the observation that pSMAD1/5 persisted in Activin A-induced ACVR1B/C dKO HOM1 cells suggested that clustering of ACVR1$^{R206H}$ by the Activin A–ACVR2A–ACVR1$^{R206H}$ complex is sufficient to lead to the activation of ACVR1$^{R206H}$, without the requirement for any upstream kinases. To test this hypothesis, we employed optogenetics to achieve light-dependent clustering of either ACVR1$^{WT}$ or ACVR1$^{R206H}$. The constructs were analogous to those published previously and comprise the light–oxygen–voltage (LOV) domain of aureochrome1

**Figure 4.  The structural, but not enzymatic, function of ACVR2A/B is essential for ACVR1$^{R206H}$ signaling.**

A   Parental HEK293T, ACVR2A KO, ACVR2B KO, ACVR2A/B dKO, HOM1, HOM1 ACVR2A KO, HOM1 ACVR2B KO, and HOM1 ACVR2A/B dKO cells were treated with Activin A or BMP2 for 1 h.

B   ACVR2A/B dKO cells were transfected with increasing concentrations of either wild-type or kinase-dead ACVR2A (ACVR2A$^{WT}$ or ACVR2A$^{KR}$, respectively) and treated with Activin A for 1 h. Parental HEK293T cells served as a control.

C   HOM1 ACVR2A/B dKO cells were transfected with increasing concentrations of either wild-type or kinase-dead ACVR2A (ACVR2A$^{WT}$ or ACVR2A$^{KR}$) and treated with Activin A for 1 h. HOM1 cells served as a control.

Data information: In all panels, Western blots of whole cell lysates were probed with the antibodies indicated. Representative experiments are shown. Par, parental HEK293T cells.

Source data are available online for this figure.

**A**

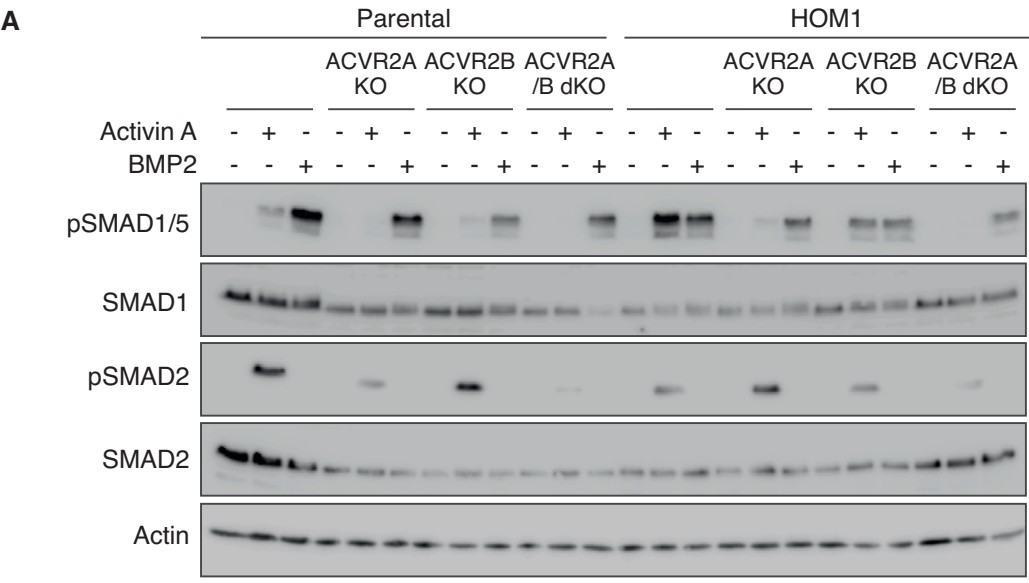

**B**

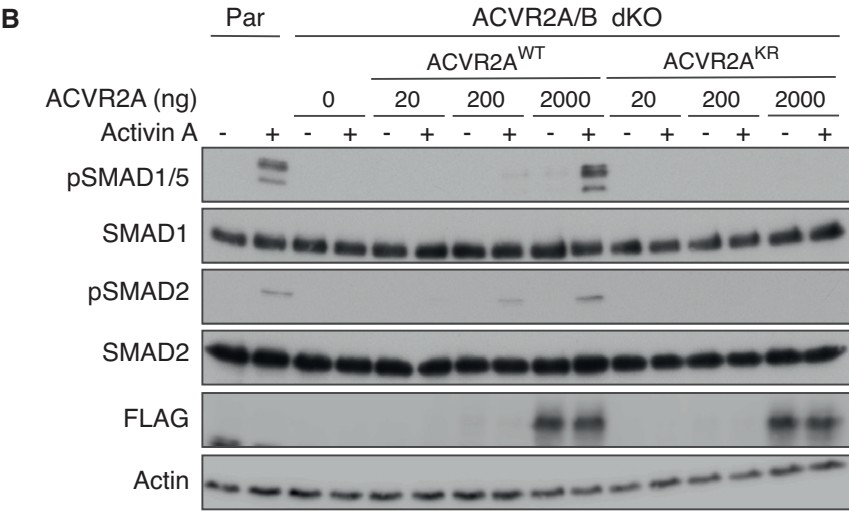

**C**

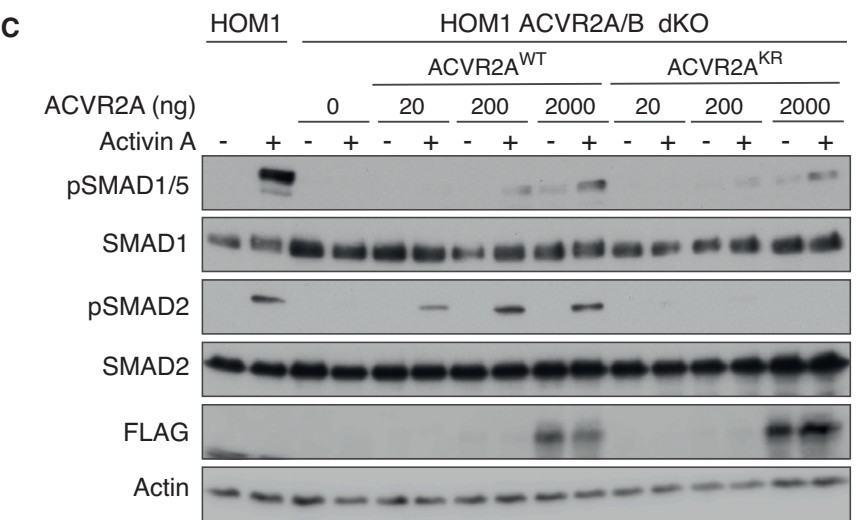

**Figure 4.**

from *Vaucheria frigida*, fused to the intracellular domains of ACVR1$^{WT}$ or ACVR1$^{R206H}$, along with an N-terminal myristoylation motif to anchor them to the plasma membrane (Ramachandran *et al,* 2018). FLAG-SMAD1 was also transfected in these experiments to increase the dynamic range of the assay. Light-dependent clustering of Opto-ACVR1$^{WT}$ was insufficient to induce SMAD1/5 phosphorylation, as previously observed (Fig 5A, lanes 3–8) (Ramachandran *et al,* 2018). However, robust light-dependent phosphorylation of FLAG-SMAD1 was observed with Opto-ACVR1$^{R206H}$, particularly when lower amounts of receptor were transfected (Fig 5A, lanes 9–14). Transfection of higher amounts of Opto-ACVR1$^{R206H}$ led to significant light-independent phosphorylation of FLAG-SMAD1, most likely by achieving supraphysiological levels of expression leading to artificial receptor clustering. Opto-ACVR1$^{R206H}$-mediated phosphorylation of FLAG-SMAD1 was completely inhibited by LDN-193189, demonstrating that the kinase activity of ACVR1$^{R206H}$ was necessary and sufficient for SMAD1/5 phosphorylation (Fig 5B, lanes 9 and 10).

The inability of Opto-ACVR1$^{WT}$ to mediate FLAG-SMAD1 phosphorylation is consistent with the requirement for an upstream kinase for activation of ACVR1$^{WT}$. Given the biochemical and genetic data placing ACVR1B/C upstream of ACVR1$^{WT}$ upon Activin A induction (Fig 3), we investigated whether constitutively active ACVR1B (Opto-ACVR1B*) could induce SMAD1/5 phosphorylation via wild-type ACVR1. As a control, we demonstrated the ability of Opto-ACVR1B* to induce phosphorylation of GFP-SMAD3 upon stimulation with blue light (Fig 5C, compare lanes 3 and 12). Strikingly, Opto-ACVR1B* also induced phosphorylation of FLAG-SMAD1 in a light-dependent manner (Fig 5C, compare lanes 7 and 16). We concluded that this was mediated via endogenous ACVR1, as it was inhibited by LDN-193189, as well as by the ACVR1B/C inhibitor SB-505124 (Fig 5C, lanes 17 and 18).

## Activin A induces ACVR2A/ACVR2B-dependent clustering in parental and HOM1 cells

From the optogenetic experiments, we concluded that receptor clustering is sufficient to induce the auto-activation of ACVR1$^{R206H}$. We hypothesized that *in vivo*, Activin A binding to ACVR2A/B provides the mechanism by which receptor clustering into higher-order structures is achieved and, therefore, developed methods to directly visualize it for endogenous receptors. First, we performed confocal microscopy and live cell imaging, applying Number and Brightness (N&B) analysis (see Methods), which has been extensively used to detect endogenous protein clustering (Digman *et al,* 2008; Nagy *et al,* 2010; Ross *et al,* 2011; Zamai *et al,* 2019). To this end, we

generated recombinant dimeric mature Activin A with an N-terminal His tag, labeled at a single amino group with the far-red fluorophore Atto 647N (His-Activin A Atto647N).

The brightness of the His-Activin A Atto647N dimer was first calibrated in solution, in the absence of cells (Fig 6A, red box in the brightness versus intensity plot (i) and color-coded (i.e., red) pixels in the map below (ii)). The B histograms (panel iii) show the frequency and distribution of the apparent brightness values (B) in the image stack; dimeric Activin A in solution has a clearly defined, bell-shaped peak (Fig 6Aiii). In the presence of either parental (Fig 6 Bi,ii) or HOM1 cells (Fig 6Ei,ii), labeled Activin A was detected as dimeric Activin A (red box and color-coded pixels), but strikingly, also as tetramers (green box and color-coded pixels) and higher-order oligomers (blue and pink boxes and color-coded pixels). Furthermore, in the presence of either parental (Fig 6Biii) or HOM1 cells (Fig 6Eiii), the B histogram distributions were right-shifted, demonstrating an increase in the frequency of pixels with higher brightness values, corresponding to higher-order oligomers (Fig 6 Biii and Eiii). These differences in Bav were reproducibly detected in multiple images (Fig 6Biv and Eiv) and were also significant across several biological replicates (Fig 6H).

To test the dependency of this clustering on the type II receptors, we used ACVR2A/B dKO parental or HOM1 cells or pre-incubated parental and HOM1 cells with ligand traps comprising the extracellular domain of human ACVR2A and ACVR2B fused to the Fc portion of human IgG$_1$ (rhACVR2A/B Fc) to inhibit Activin A binding to endogenous ACVR2A/B (Fig EV4A). When labeled Activin A was added to ACVR2A/B dKO cells, or cells were pre-incubated with rhACVR2A/B Fc, the occurrence of tetramers and higher-order oligomers was significantly reduced in both parental and HOM1 cells (Fig 6C, D, F and G, respectively). This was evident in the B histograms, which showed a left-shifted distribution, demonstrating a decrease in the number of pixels with higher brightness values. In the case of the ligand traps, the Bav values were reduced to the value of dimeric Activin A in solution (Fig 6D, G and H), while in ACVR2A/B dKO cells, the average brightness was slightly higher than that of dimeric Activin A, although the difference was not significant (Fig 6C, F and H). Thus, in both parental and HOM1 cells, Activin A-induced receptor clustering requires ACVR2A/B.

We next investigated whether Activin A also triggers receptor clustering in disease relevant models using the DIPG cell lines ICR-B169 and HSJD-DIPG-007. In both cell lines, labeled Activin A clustered into tetramers and higher-order oligomers, compared with Activin A in the absence of cells (Fig EV4B, Ci,ii and Ei,ii, green, blue and pink boxes, respectively) that was reflected in right-shifted histograms and Bav values in multiple images (Fig EV4Ciii,iv and

---

**Figure 5. Receptor clustering is sufficient for ACVR1$^{R206H}$ signaling.**

A, B   NIH-3T3 cells were either untransfected or transfected with FLAG-SMAD1 together with increasing (doubling) amounts of Opto-ACVR1$^{WT}$ or Opto-ACVR1$^{R206H}$ (5–20 ng). Post-transfection, cells were either kept in the dark or exposed to blue light for 1 h. In (B), cells were treated in the presence or absence of 1 μM LDN-193189.

C       NIH-3T3 cells were either untransfected or transfected with either FLAG-SMAD1 or GFP-SMAD3 together with Opto-ACVR1B*. Post-transfection, cells were either kept in the dark or exposed to blue light for 1 h, in the presence or absence of 50 μM SB-505124 or 1 μM LDN-193189.

Data information: In all panels, Western blots of whole cell lysates were probed with antibodies against pSMAD1/5 (that detected both endogenous and pFLAG-SMAD1), SMAD1 (which detects both endogenous and FLAG-SMAD1), HA (which detects the Opto-receptors), pSMAD3 (detects pGFP-SMAD3), SMAD3 (which detects GFP-SMAD3), and either tubulin or actin as the loading controls. Lane numbers are given below each blot for clarity.

Source data are available online for this figure.

**A**

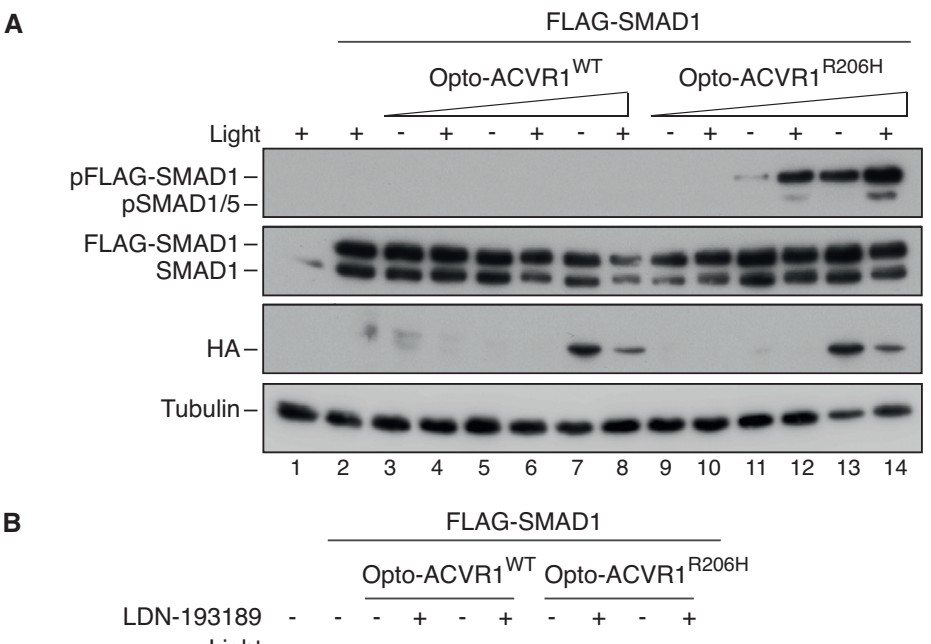

**B**

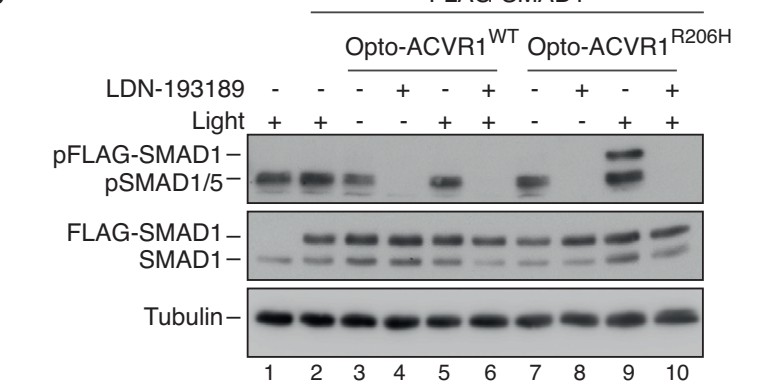

**C**

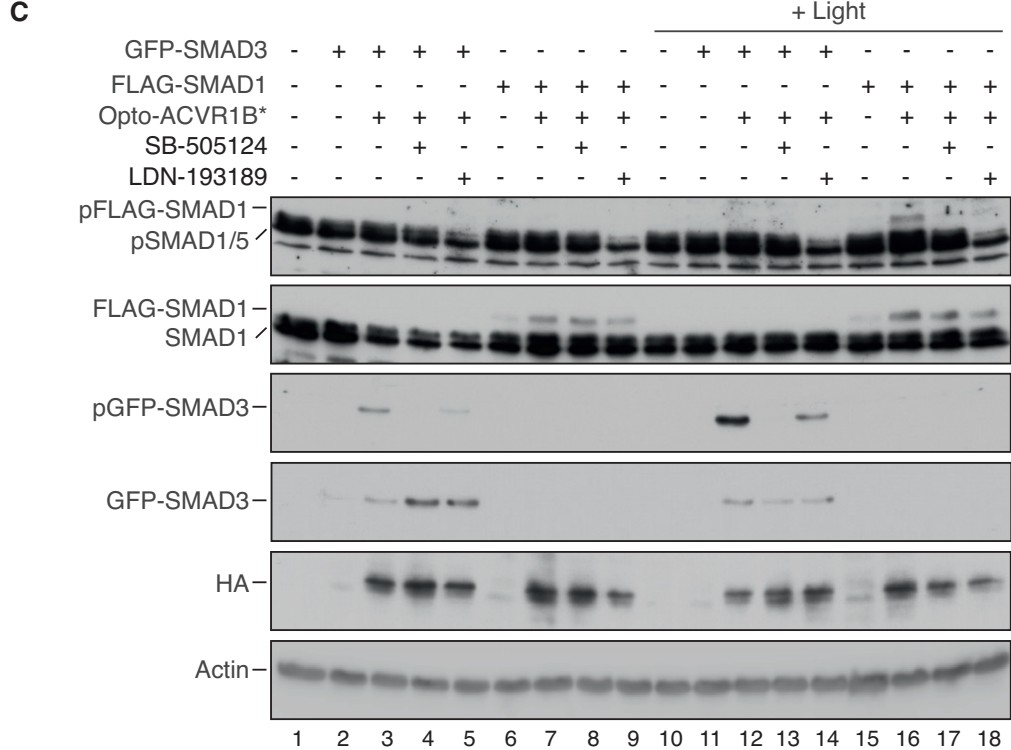

**Figure 5.**

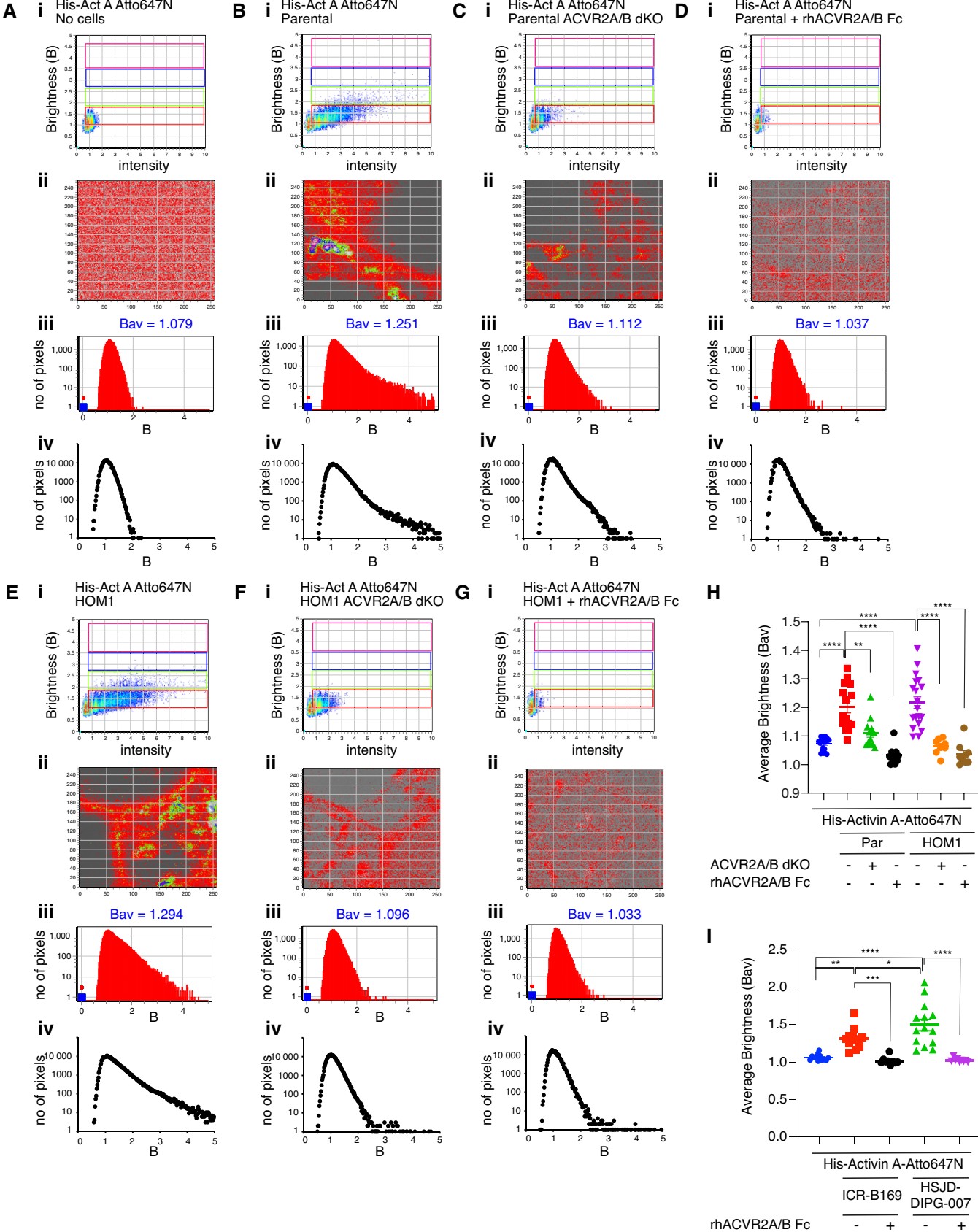

**Figure 6.**

**Figure 6. ACVR2A/B are required for Activin clustering.**

A–G   His-Activin A-Atto647N was imaged free in solution with no cells (A) or with parental HEK293T cells (B), ACVR2A/B dKO cells (C), parental HEK293T cells pre-incubated with 1.5 µg/ml of rhACVR2A Fc and 15 µg/ml of rhACVR2B Fc (D), HOM1 cells (E), HOM1 ACVR2A/B dKO cells (F), and HOM1 cells pre-incubated with 1.5 µg/ml of rhACVR2A Fc and 15 µg/ml of rhACVR2B Fc (G). Number and Brightness (N&B) analysis was performed on raster-scanned images, and a representative experiment is shown. Panel i, Brightness (B) *us* intensity 2D plots. Red rectangular box, pixels that contribute to brightness of free, dimeric His-Activin A-Atto647N in solution. Green rectangular box, pixels that contribute to His-Activin A-Atto647N clustered into putative tetramers. Blue and pink rectangular boxes, pixels that contribute to His-Activin A-Atto647N in higher-order clusters. Panel ii, Selection map of the image captured reconstructed with the pixels in red, green, blue, and pink rectangular boxes shown in the upper panel (i). Panel iii, B histograms, frequency distribution of the number of pixels across various brightness of His-Activin A-Atto647N of the selected image. Bav indicates the average brightness for each image. Panel iv, cumulative B histograms from a representative experiment with 4 independent regions of interest.

H    Average brightness of His-Activin A-Atto647N in conditions (A–G). Mean ± SEM are plotted. Each point represents one analyzed image, collected from 2–3 biological replicates.

I    Average brightness of His-Activin A-Atto647N in ICR-B169 and HSJD-DIPG-007 cells alone or with pre-incubation with 1.5 µg/ml of rhACVR2A Fc and 15 µg/ml of rhACVR2B Fc. Source data are in Fig EV4B–F. Mean ± SEM are plotted. Each point represents one analyzed image, collected from 2–4 biological replicates.

Data information: In (H and I), the *P*-values are from one-way ANOVA with Tukey's *post hoc* correction. *$P < 0.05$; **$P < 0.01$; ***$P < 0.001$; and ****$P < 0.0001$.
Source data are available online for this figure.

Eiii,iv) across several biological replicates (Fig 6I). Interestingly, HSJD-DIPG-007 cells demonstrated higher Activin A-induced clustering compared to ICR-B169 cells (Figs EV4E and C, and 6I). In both cell lines, pre-incubation with ligand traps significantly reduced higher-order receptor oligomerization with a concomitant decrease in Bav values (Figs EV4D and F, and 6I). Thus, in ACVR1$^{WT}$ and ACVR1$^{R206H}$ glioma cells, Activin A induces receptor clustering in a manner dependent on the presence of the type II receptors.

Finally, to demonstrate receptor clustering more visually, we turned to total internal reflection fluorescence (TIRF) microscopy of artificial planar lipid bilayers and specifically focused on the HOM1 cells. This method has been used extensively to study T- and B-cell receptor clustering in the immunological synapse, as well as tyrosine kinase receptor clustering (Grakoui *et al*, 1999; Tolar *et al*, 2009; Salaita *et al*, 2010). We attached His-Activin A Atto647N to nickel-containing phospholipids in an artificial mobile lipid bilayer on a glass coverslip. The His-Activin A-Atto647N was anchored, but could diffuse freely within the bilayer in the XY plane. Upon plating of cells on the bilayer, Activin A-induced receptor clustering is visualized as a localized reduction in mobility of the fluorescent Activin A that is evident as a focused increase in fluorescence intensity.

As a technical control to demonstrate the mobility of the bilayer, we used 4-Hydroxy-3-iodo-5-nitrophenyl)acetyl (NIP)$_1$-H12-Hylight647 ((NIP)$_1$-Hylight647), a fluorescently labeled His-tagged small molecule B-cell antigen, previously used on lipid bilayers (Tolar *et al*, 2009), that does not bind Activin receptors (Movie EV1). Plating HOM1 cells on this bilayer revealed that the cells spread, but (NIP)$_1$-Hylight647 remained dispersed and freely diffusible, with only occasional small clusters (Fig 7A and D; Movie EV2). Time-lapse TIRF imaging of His-Activin A-Atto647N on the bilayer for 20 s before the addition of cells demonstrated that it too was mobile and dispersed (Movie EV3; Fig 7C). Importantly, upon plating HOM1 cells on the lipid bilayer containing His-Activin A-Atto647N, time-lapse TIRF imaging for 20 min showed that shortly after cell spreading, prominent fluorescent clusters appeared (Fig 7 B; Movie EV4). The first clusters were already apparent at the 0 min time point, taken approximately 5 min after cell plating, but became more intense and numerous over time (Fig 7B). Quantification of cluster intensities showed that areas of the Activin A-containing bilayer in contact with cells (as defined by the brightfield images) had significantly brighter clusters, and the brightness increased over time, compared to areas of the bilayer without cells (Fig 7D).

As a further control to demonstrate that the clustering was not caused by nonspecific aggregation of the Atto647N fluorophore which is known to be hydrophobic, we conjugated His-tagged Activin to another far-red fluorophore (CF640R), that is hydrophilic and less prone to aggregation (Zanetti-Domingues *et al*, 2013). The results were consistent with what we had shown for Activin A-Atto647N. Time-lapse TIRF imaging of His-Activin-A-CF640R on the bilayer for 20 s (Movie EV5) or 7.5 min (Fig EV5A; Movie EV6) demonstrated its mobility, and upon plating the HOM1 cells on the His-Activin-A-CF640R-containing lipid bilayers, clusters of labeled Activin A appeared and increased in intensity over time (Fig EV5B, C and D; Movie EV7). Thus, we conclude that Activin A binding to receptors in HOM1 cells induces substantial receptor clustering.

**Figure 7. TIRF imaging of Activin A-induced clustering on artificial lipid bilayers.**

A, B   HEK293T HOM1 cells were plated on a planar lipid bilayer containing (NIP)$_1$-H12-Hylight647 (A) or His-Activin A-Atto647N (B) and an automated time-lapse was started, where 2–4 positions within a well were imaged every 40 s during a 27-min time period; 3 time points from one position are shown. TIRF, upper panels and brightfield, lower panels. The cluster intensities from all positions were quantified at 6 time points indicated (right panel) and plotted as individual cluster values. The horizontal gray bar denotes the mean. Cells + bilayer denotes bilayer areas under the cells, as identified from the brightfield images, and bilayer denotes areas where no cells were present. A representative experiment from 2 biological replicates quantified is shown. Scale bar 10 µm. In (B), the region under cell 2 is shown enlarged in the top right-hand corner.

C    Mobile His-Activin A-Atto647N on a planar lipid bilayer was imaged for a 20 s period; 3 time points are shown. Scale bar 10 µm.

D    Mean cluster intensity per cell over time. Quantifications are the mean cluster intensities at 5 time points of 35 cells (NIP + cells), 99 cells (Activin A + cells), and 28 bilayer areas ± SEM, from a representative experiment imaged as in (A and B). The *P*-values are from *F*-test. ns, not significant and **$P < 0.01$.

Source data are available online for this figure.

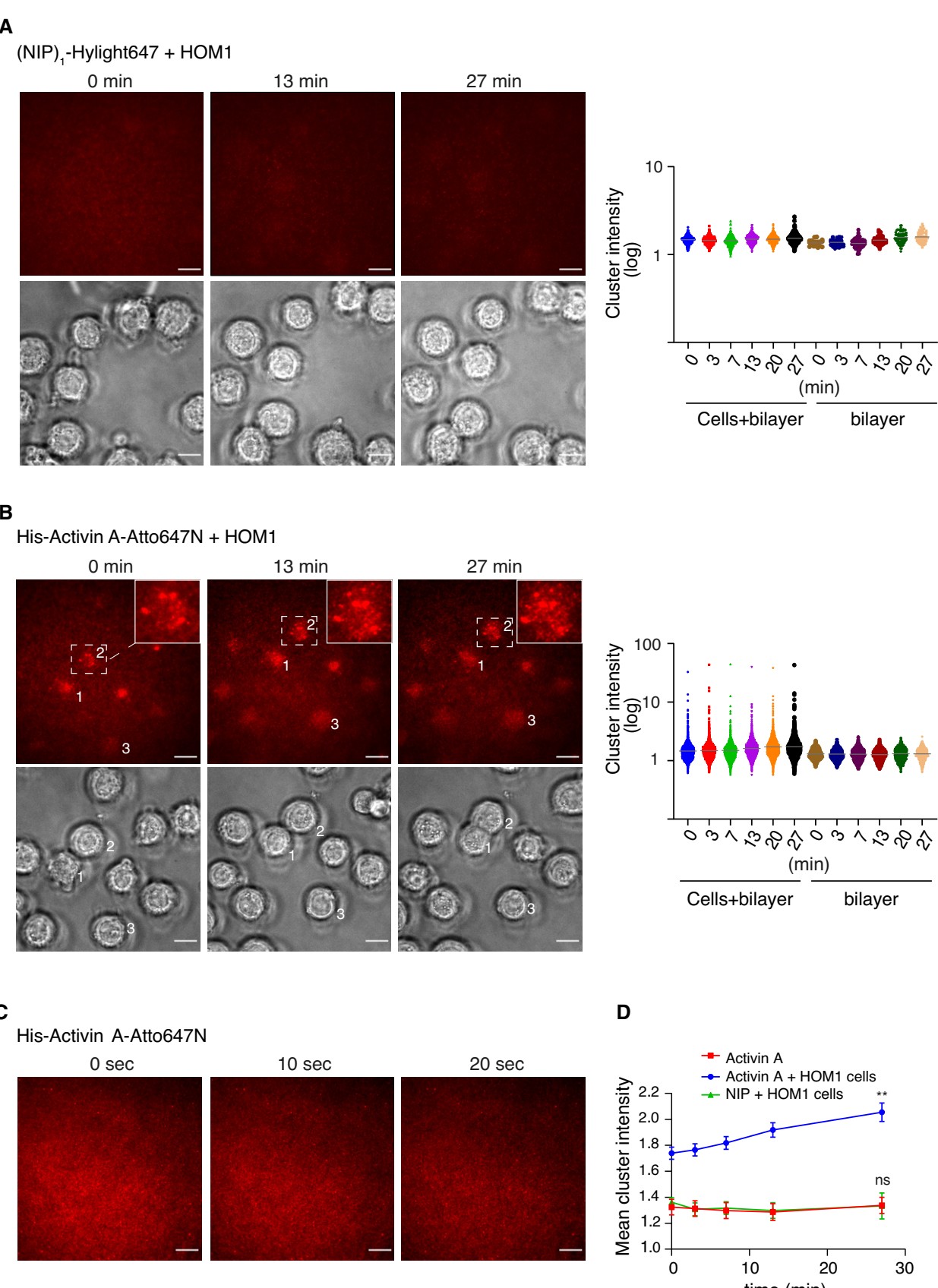

**Figure 7.**

## Discussion

Here, we demonstrate the mechanism whereby ACVR1$^{WT}$ and ACVR1$^{R206H}$ are activated by Activin A to induce the SMAD1/5 signaling pathway. We have shown that for ACVR1$^{WT}$, this occurs by formation of a canonical Activin A–ACVR2A/B–ACVR1B/C complex, whereupon the activated ACVR1B/C phosphorylates and activates ACVR1. This is analogous to the mechanism we previously showed underpins the activation of SMAD1/5 by TGF-β (Ramachandran *et al*, 2018). For both ligands, the fundamental new concept is the ability of one activated type I receptor to phosphorylate and activate another type I receptor. For ACVR1$^{R206H}$, we have shown that this mechanism also operates, particularly at low Activin A levels. However, for the mutant receptor the predominant mechanism involves direct clustering of ACVR2A/B–ACVR1$^{R206H}$ complexes by Activin A. In this case, the role of the type II receptor is structural, as its kinase activity is not required. This view is supported by opto-genetics experiments, showing that light-induced dimerization of opto-ACVR1$^{R206H}$ is sufficient to induce a robust phosphorylation of SMAD1/5. For cells containing either ACVR1$^{WT}$ or ACVR1$^{R206H}$, including glioma cells from DIPG patients, we demonstrate that Activin A induces formation of higher-order clusters, as determined by N&B analysis of live confocal images and as directly visualized by the clustering of fluorescent Activin A on synthetic lipid bilayers. Thus, ACVR1$^{R206H}$ establishes a new paradigm in TGF-β family signaling, where only its kinase activity is required for signaling.

### Ligand selectivity of ACVR1$^{WT}$ versus ACVR1$^{R206H}$

Here, we have demonstrated that Activin A induces pSMAD1/5 via ACVR1$^{WT}$ and ACVR1$^{R206H}$, and a recent study in a multiple myeloma cell line also demonstrated this activity in ACVR1$^{WT}$ cells (Olsen *et al*, 2020). We note that previous studies have readily detected Activin A-induced pSMAD1/5 in ACVR1$^{WT}$ cells, but this has gone unacknowledged (Hatsell *et al*, 2015; Hino *et al*, 2015; Aykul *et al*, 2020), leading to the conclusion that the R206H mutation conferred a neofunction with respect to Activin induction. Activin A was suggested to form a non-signaling complex with ACVR2A/B and ACVR1, but a functional signaling complex when ACVR1 was mutated (Hatsell *et al*, 2015; Aykul *et al*, 2020). One possible reason for this interpretation could be the assays used. The BRE-Luciferase reporter has been widely used in other studies as a readout of Activin A-induced SMAD1/5 signaling. However, the assay requires sufficient time for luciferase levels to accumulate and while we have shown previously that it is an excellent readout of BMP activity via pSMAD1/5, it does not respond to TGF-β-induced SMAD1/5 signaling, as it is potently inhibited by the SMAD3 arm of the TGF-β pathway (Daly *et al*, 2008; Grönroos *et al*, 2012). As Activin A also stimulates the SMAD3 pathway, we have avoided this reporter assay as a readout for Activin-induced pSMAD1/5 signaling and instead have predominantly used phosphorylation of SMAD1/5 as a direct pathway readout. We have also shown by qPCR that the extent of pSMAD1/5 induced by both ACVR1$^{WT}$ and ACVR1$^{R206H}$ is mirrored in the induction of SMAD1/5 target genes, demonstrating that in all cases it is functional. From our data, we conclude that the R206H mutation does not confer a neofunction on ACVR1. Instead, pSMAD1/5 in response to Activin A is a widespread phenomenon in cells that respond to Activin A, reflected at both the protein level

and by target gene expression. ACVR1$^{R206H}$ enables the phosphorylation of SMAD1/5 in response to Activin A via a distinct mechanism from wild type, requiring only receptor clustering via the type II receptor.

Interestingly, using MEFs from an FOP mouse model, we have also shown that TGF-β can activate ACVR1$^{R206H}$, but not ACVR1$^{WT}$, via this novel mechanism, although presumably TGFBR2 is the relevant type II receptor in this case. This result suggests that TGF-β might also be a relevant ligand *in vivo* for activating the pSMAD1/5 pathway via ACVR1$^{R206H}$. Of note, we do not see a clear differential between the ability of ACVR1$^{WT}$ and ACVR1$^{R206H}$ to induce pSMAD1/5 when cells are treated with Activin B. We have not delved into the underlying mechanism in detail, but it seems to reflect a higher efficiency of signaling via ACVR1$^{WT}$ in response to Activin B compared with Activin A. It is possible that the Activin type I receptor plays a role in this, as Activin A has a preference for ACVR1B, while Activin B can signal through ACVR1B or ACVR1C (Goebel *et al*, 2019).

### ACVR1$^{R206H}$ has higher catalytic activity than the wild-type receptor, and can be activated without the requirement for upstream kinases

The tight time course of Activin A induction performed for wild-type versus mutant ACVR1 revealed that the kinetics of SMAD1/5 phosphorylation and thus receptor activation efficiency is different, with the mutant receptor responding much more rapidly than the wild type. This is consistent with previous structural work showing that the R206H mutation destabilizes the inactive state of the receptor (Chaikuad *et al*, 2012). In wild-type ACVR1, R206, which is part of the GS domain, is important for the interaction of the GS domain with the N-lobe of the kinase domain, hence maintaining the inactive state. The His mutation destabilizes these interactions and also reduces the binding of FKBP1A. Thus, the mutant receptor is in a conformation poised for activation (Shen *et al*, 2009; Chaikuad *et al*, 2012).

We hypothesize that the partially active conformation of ACVR1$^{R206H}$ explains the lack of requirement for an upstream kinase for activation. Our optogenetic experiments with this mutant receptor alone demonstrate that dimerization is sufficient for robust activation. We propose that in these homomeric complexes, the kinase domains phosphorylate and activate each other, allowing SMAD1/5 binding and phosphorylation. A similar process occurs in the type II receptor rescue experiments, when kinase-dead ACVR2A is used. In this case, we propose that Activin A binding to ACVR2A$^{KR}$ and ACVR1$^{R206H}$ brings the mutated type I receptors in close proximity and hence triggers their auto-activation. In the case of ACVR1$^{WT}$, Activin-induced SMAD1/5 phosphorylation is completely inhibited by the ACVR1B/C inhibitor, SB-431542, demonstrating that ACVR1 activation by Activin A is indirect and mediated via activated ACVR1B/C. We have additionally confirmed this with optogenetic and knockout experiments.

Our data and the work of others raise the interesting question of why for ACVR1$^{WT}$, an Activin A–ACVR2A/B–ACVR1 complex does not lead to signaling, whereas ACVR1$^{WT}$ can be directly activated in a conventional ligand–type II–ACVR1 complex by BMPs (this work; Macías-Silva *et al*, 1998; Hatsell *et al*, 2015; Aykul *et al*, 2020). This observation suggests that type II receptors cannot phosphorylate

and activate ACVR1 when brought into close proximity by Activin A, but can when the ligand is a BMP. This question has not been answered, but we can speculate that the reason may lie in the different affinity of Activin for the receptor complex, compared with the BMPs, or possibly it is due to the specific type II receptor in the complex (Hinck *et al*, 2016). Notably, ACVR2A and ACVR2B bind Activin 100–1,000 times more avidly than they do BMPs, and BMPR2 binds BMPs, but not Activins (Hinck *et al*, 2016). Alternatively, the orientation of the kinase domains in the different complexes may be different for Activin–receptor complexes versus BMP–receptor complexes. Indeed, a recent structure of a GDF11–ACVR2B–TGFBR1 complex, which is thought to mirror an Activin-containing complex, has a distinct architecture and mode of receptor binding compared with either BMP- or TGF-β-containing receptor complexes (Goebel *et al*, 2019).

### Activin A-induced phosphorylation of SMAD1/5 is transient

Our data demonstrate that Activin A-induced phosphorylation of SMAD1/5 is attenuated over time in both wild-type and mutant cells, with the duration of signaling slightly prolonged in the HOM1 cells compared to parental. We have shown that in both cases, prolonged signaling requires the constant presence of extracellular ligand, as a follistatin chase after a 1-h Activin A stimulation curtails signaling in both parental and HOM1 cells. This indicates that signaling requires constant activation of new receptors at the cell surface. TGF-β family signal attenuation is driven by receptor degradation mediated via the ESCRT machinery (Miller *et al*, 2018; Miller *et al*, 2019). We show here that pathway attenuation requires the Activin-induced SMAD2/3 pathway, at least partly via its ability to induce expression of SMAD6 and SMAD7. We do not yet understand the molecular mechanism underlying the slightly prolonged Activin A-induced pSMAD1/5 signaling in HOM1 cells, compared with parental cells, but it could reflect a decreased degradation rate for ACVR1$^{R206H}$ versus ACVR1$^{WT}$. We speculate that this could result from a difference in binding affinity of the receptors for SMAD6/7 and/or a difference in efficiency of receptor ubiquitination that is required for receptor trafficking to the lysosomes (Miller *et al*, 2018; Miller *et al*, 2019).

### Activin A stimulation leads to receptor clustering and formation of higher-order receptor complexes

For receptor tyrosine kinases, it has been well established that dimerization and higher-order hetero-oligomerization are required for efficient receptor activation (Schlessinger, 2000), but the formation of higher-order ligand–receptor complexes has not previously been demonstrated for TGF-β family receptors. Here, we utilize two live-imaging methods for studying Activin A-induced receptor clustering and demonstrate the rapid formation of higher-order complexes in both parental and HOM1 cells. The formation of these complexes does not depend on the mutational status of ACVR1, but is entirely dependent on the presence of ACVR2A/B. We propose that formation of higher-order receptor complexes enables the type I receptors to activate each other, either in parental cells when activated ACVR1B/C phosphorylate and activate ACVR1 or in the mutant cells where ACVR1$^{R206H}$ receptors activate each other. We note that this might allow the mutant ACVR1 to initiate signaling in

additional cell types lacking ACVR1B/C, and when Activin A levels are high, such as during inflammation in the context of FOP (Alessi Wolken *et al*, 2018). We concluded previously that TGF-β also induces formation of higher-order receptor complexes (Ramachandran *et al*, 2018). It will be important in the future to demonstrate this directly using the approaches pioneered here.

## Materials and Methods

### Cell lines

All cell lines were maintained at 37°C and 10% $CO_2$, except the DIPG cell lines which were kept at 5% $CO_2$, and have been certified mycoplasma negative by the Francis Crick Institute Cell Services. HEK293T cells were obtained from the Francis Crick Institute Cell Services, and NIH-3T3 cells were obtained from Dr Richard Treisman (Francis Crick Institute, London). HEK293T cells were authenticated using short tandem repeat (STR) profiling, while NIH-3T3 cells had species confirmation at the Francis Crick Institute Cell Services. Mouse embryonic fibroblasts (MEFs) from *Acvr1*$^{+/+}$ and *Acvr1*$^{+/R206H}$ mice were isolated as previously described (Xu, 2005; Culbert *et al*, 2014). The ICR-B169 and HSJD-DIPG-007 cell lines derived from DIPG patient gliomas (Carvalho *et al*, 2019) were obtained from Professor Chris Jones (ICR, London) in whose laboratory STR profiling had been performed. Patients consented to inclusion in a local IRB/Research Ethics Committee-approved project at the respective institutions (Hospital San Joan de Deu, Barcelona for HSJD-DIPG-007; ICR, London for ICR-B169). These ethical approvals cover the use of the cell lines in collaborations beyond the institute the cell lines were established in. For further details about these lines, see https://www.crukchildrensbraintumourcentre.org/research/resources/cell-line-repository/. They were cultured as described in Taylor *et al* (2014a). In brief, cells were cultured under adherent stem cell conditions in flasks coated with Laminin (R&D systems) in neurobasal-A medium (Invitrogen), supplemented with B-27 (Invitrogen) and the growth factors epidermal growth factor (EGF), basal fibroblast growth factor (bFGF), platelet-derived growth factor (PDGF)-AA, and PDGF-BB (all from Shenandoah Biotech).

HEK293T, human embryonic kidney cells, transformed with SV40 T-antigen, maintained in Dulbecco's modified Eagle's medium (DMEM) + 10% fetal calf serum (FCS) + 1% penicillin/streptomycin (Pen/Strep).

NIH-3T3 cells, mouse embryonic fibroblasts, maintained in DMEM + 10% FCS + 1% Pen/Strep.

MEFs (*Acvr1*$^{+/+}$ and *Acvr1*$^{+/R206H}$), mouse embryonic fibroblasts from *Acvr1*$^{+/+}$ and *Acvr1*$^{+/R206H}$ mice, maintained in DMEM + 10% FCS + 1% Pen/Strep.

The following were all derived from HEK293T cells and were all maintained in DMEM + 10% FCS + 1% Pen/Strep. Sequences of the genome-edited alleles are given in Appendix Supplementary Methods.

HEK293T parental clone, single cell clone isolated and expanded from the original HEK293T pool that responded to Activin A and BMP2 like the original pool and in which all genome editing was performed.

HET, HEK293T clone in which one allele was ACVR1$^{WT}$ and the other ACVR1$^{R206H}$.

HOM1, HEK293T clone in which only ACVR1^R206H was detected.

HOM2, HEK293T clone in which only ACVR1^R206H was detected.

ACVR1 KO1, HEK293T clone in which ACVR1 had frameshift mutations leading to predicted protein truncations and loss of function.

ACVR1 KO2, HEK293T clone in which ACVR1 had frameshift mutations leading to predicted protein truncations and loss of function.

S2/3 dKO1, HEK293T clone in which both SMAD2 and SMAD3 had frameshift mutations leading to predicted protein truncations and loss of function.

S2/3 dKO2, HEK293T clone in which both SMAD2 and SMAD3 had frameshift mutations leading to predicted protein truncations and loss of function.

S6 KO1, HEK293T clone in which SMAD6 had frameshift mutations leading to predicted protein truncations and loss of function.

S6 KO2, HEK293T clone in which SMAD6 had frameshift mutations leading to predicted protein truncations and loss of function.

S7 KO1, HEK293T clone in which SMAD7 had frameshift mutations leading to predicted protein truncations and loss of function.

S7 KO2, HEK293T clone in which SMAD7 had frameshift mutations leading to predicted protein truncations and loss of function.

S6/7 dKO1, HEK293T clone in which both SMAD6 and SMAD7 had frameshift mutations leading to predicted protein truncations and loss of function.

S6/7 dKO2, HEK293T clone in which both SMAD6 and SMAD7 had frameshift mutations leading to predicted protein truncations and loss of function.

ACVR1B/C dKO, HEK293T clone in which both ACVR1B and ACVR1C had frameshift mutations leading to predicted protein truncations and loss of function.

HOM1 ACVR1B/C dKO, HEK293T HOM1 in which both ACVR1B and ACVR1C had frameshift mutations leading to predicted protein truncations and loss of function.

ACVR2A KO, HEK293T clone in which ACVR2A had frameshift mutations leading to predicted protein truncations and loss of function.

ACVR2B KO, HEK293T clone in which ACVR2B had frameshift mutations leading to predicted protein truncations and loss of function.

ACVR2A/B dKO, HEK293T clone in which both ACVR2A and ACVR2B had frameshift mutations leading to predicted protein truncations and loss of function.

HOM1 ACVR2A KO, HEK293T HOM1 in which ACVR2A had frameshift mutations leading to predicted protein truncations and loss of function.

HOM1 ACVR2B KO, HEK293T HOM1 in which ACVR2B had frameshift mutations leading to predicted protein truncations and loss of function.

HOM1 ACVR2A/B dKO, HEK293T HOM1 in which both ACVR2A and ACVR2B had frameshift mutations leading to predicted protein truncations and loss of function.

HSJD-DIPG-007, primary culture derived from DIPG patient with ACVR1 R206H and K27M H3F3A mutations.

ICR-B169, primary culture derived from DIPG patient with wild-type ACVR1 and K27M H3F3A mutation.

## Ligands, chemicals and cell culture

Activin A (PeproTech) and Activin B (PeproTech) were reconstituted in phosphate buffered saline (PBS) supplemented with 0.1% bovine serum albumin (BSA). TGF-β (PeproTech), BMP4 (PeproTech), BMP7 (PeproTech), the BMP4-BMP7 heterodimer (BMP4/7) (R&D Systems), and BMP2 (R&D Systems) were reconstituted in 4.4 mM HCl supplemented with 0.1% BSA. Unless otherwise noted, all ligands were used at 20 ng/ml except TGF-β, which was used at 2 ng/ml. Growth factors (bFGF, EGF, PDGF-AA, PDGF-BB) used in the tumor stem cell medium were reconstituted in stem cell medium just before use, and used at 20 ng/ml (bFGF, EGF) or 10 ng/ml (PDGF-AA, PDGF-BB) final concentration. Cells were treated with the indicated ligands for the times indicated in the figures. SB-431542 (Tocris, UK) was used at a final concentration of 10 μM, SB-505124 (Tocris) was used at a final concentration of 50 μM, and LDN-193189 (a gift from Paul Yu) was used at a final concentration of 1 μM. For Activin A blocking experiments, follistatin 288 (Sigma) was reconstituted in PBS and used at a final concentration of 500 ng/ml. Recombinant human (rh) ACVR2A Fc chimera protein (R&D Systems) and rhACVR2B Fc chimera (R&D Systems) were reconstituted in PBS and used according to the manufacturer's instructions. Prior to ligand stimulation, HEK293T cells and NIH-3T3 cells were starved overnight in DMEM containing 0.5% FCS. For the MEFs, cells were starved in OptiMEM with 10 μM SB-431542 overnight and then the SB-431542 was washed out prior to ligand stimulation.

## Preparation of fluorescently tagged recombinant Activin

Mature Activin A (amino acids 311-426, UniProt: P08476) with N-terminal hexa-His and StrepTags (tag sequence: MNTIHHHHHHNTS-SATMNSWSHPQFEKSM) was expressed in *E. coli*, refolded, and purified to homogeneity. Washed inclusion bodies were first solubilized in 20 mM Tris(2-carboxyethyl)phosphine (TCEP), 6 M Guanidinium HCl, 50 mM Tris–HCl pH 8.0, and 2 mM EDTA. Solubilized protein was buffer-exchanged to 6 M urea and 20 mM HCl and then diluted 10× into refolding solution of 1 M pyridinyl propyl sulfonate, 50 mM Tris–HCl, 50 mM ethanolamine, 1 mM EDTA, 2 mM cysteine, and 0.2 mM cystine, pH 8.0. Refolded protein was purified by a combination of cation exchange and reversed phase chromatography. Pure, dimeric Activin A was dried under vacuum and stored in aliquots. For labeling, Activin A was solubilized in 10 mM HCl and diluted to 1 mg/ml concentration in a solution containing 50 mM Hepes pH 7.4, 30% acetonitrile. NHS-activated fluorophores (Atto 647N (Sigma-Aldrich) or CF640R (Insight Biotechnology)), both 10 mM stock solutions in DMSO, were added to the protein in 2–8× molar excess and incubated for 2–3 h at room temperature. The labeling reaction was terminated by addition of 0.1% trifluoroacetic acid. The sample was loaded onto an ACE C4 reversed phase column and eluted by acetonitrile gradient, following absorbance at maximum absorption wavelength of the fluorophore. Labeled protein fractions were analyzed by SDS–PAGE and degree of labeling determined spectrophotometrically as the ratio of fluorophore concentration and protein concentration. Fractions with lowest degree of labeling (less than two labels per dimeric Activin A) were used for the experiments. Labeled protein was dried under vacuum and stored in single use aliquots in black tubes at −80°C.

## Knock-in of ACVR1^R206H at the endogenous locus

From wild-type HEK293T cells, a parental clone was selected whose response to Activin A and BMP2 phenocopied the wild-type pool.

Guide RNAs (gRNAs) for CRISPR/Cas9-mediated genome editing were expressed from the pSpCas9(BB)-2A-GFP (PX458) plasmid (Ran *et al*, 2013). All gRNA sequences and cloning oligonucleotides are listed in Table EV1. A gRNA immediately downstream of codon 206 was chosen for the knock-in of the ACVR1$^{R206H}$ point mutation (homology-directed repair gRNA) and cloned into PX458. A 100 nucleotide single-stranded DNA (ssDNA) of HPLC purity was provided as the repair template (Table EV1). The ssDNA was modified by phosphorothioate bonds between the first two and last two nucleotides at the 5' and 3' ends, respectively. The repair template alters codon 205 (nucleotides 613-615) from GCT → GCG (Ala → Ala) and codon 206 (nucleotide 616-618) from CGC → CAT (Arg → His). The silent mutation of codon 205 was introduced to increase the specificity of the downstream screening primer. The mutation of codon 206 additionally destroys the PAM sequence for the gRNA. Four million HEK293T cells were plated in a 10-cm dish in a total volume of 10 ml media. The following day, a transfection mix was made with 7.5 μg of pX458-gRNA plasmid, 1 μl of 100 μM ssDNA repair template, and 30 μl of FuGENE HD in a total of 500 μl OptiMEM. Complexes were allowed to form at room temperature for 15 min and then added to HEK293T cells. Forty-eight hour post-transfection, HEK293T cells were single cell sorted for GFP expression into 96-well plates. Ten days post-sorting, clones were consolidated and replica plated into two 96-well plates—one plate was used as the cell stock, while the other was used for screening. Once confluent, cells were lysed with the QuickExtract DNA Extraction Solution (Lucigen) according to the manufacturer's instructions. The same universal reverse primer (Table EV1) was used to screen all clones. Two different forward primers were used: either the ACVR1$^{WT}$ forward primer to detect the ACVR1$^{WT}$ allele or the ACVR1$^{R206H}$ forward primer to detect the ACVR1$^{R206H}$ knock-in allele. Clones that were positive for the ACVR1$^{R206H}$ knock-in by PCR were selected for further analysis and verified by Sanger sequencing. Of note, during the CRISPR/Cas9-mediated genome editing, homology-directed repair did not always occur leading to the generation of ACVR1 knockout clones that were also used in this study.

## CRISPR/Cas9-mediated Gene Knockouts

Guide RNAs targeting ACVR1B, ACVR1C, ACVR2A, ACVR2B, SMAD2, SMAD3, SMAD6, and SMAD7 (Table EV1) were cloned into PX458. gRNAs were designed to target the intracellular domain of the receptors just upstream of the GS domain (type I receptors) or kinase domain (type II receptors). SMAD knockout clones were generated in the HEK293T parental clone only. All receptor knockouts were generated in both HEK293T parental and HEK293T HOM1 cells. Transfections and cell sorting were performed as described above, with the omission of the ssDNA repair template. Clones were verified by Sanger sequencing.

## Transient transfection of ACVR2A

Wild-type ACVR2A (ACVR2A$^{WT}$) and kinase-dead ACVR2A in which codon 219 was mutated from lysine to arginine (ACVR2A$^{K219R}$, referred to as ACVR2A$^{KR}$) (Table EV1) were cloned into pcDNA3.1 Hygro (+) (Thermo Fisher), both with a C-terminal FLAG tag. Plasmids were transiently transfected using FuGENE HD into both ACVR2A/B dKO and HOM1 ACVR2A/B dKO cells. Cells

were transfected with 2 μg, 200 ng, or 20 ng of the appropriate plasmids in a 6-well format; where less than 2 μg of ACVR2A$^{WT/KR}$ plasmid was transfected, the quantity was made up to 2 μg with empty pcDNA3.1 Hygro (+). Twenty-four hours post-transfection, cells were starved overnight. The following days, cells were stimulated with 20 ng/ml Activin A.

## Generation of Opto-Receptors and LED light photoactivation

Opto-ACVR1$^{WT}$ was previously described (Ramachandran *et al*, 2018), and site-directed mutagenesis was performed to generate Opto-ACVR1$^{R206H}$ by mutating codon 206 from CGC to CAC (Table EV1). Both receptors have a C-terminal HA tag. Opto-ACVR1B* was generated in pCS2+ as previously described (Ramachandran *et al*, 2018) and included an N-terminal myristoylation domain, the intracellular portion (residues 150–505) of the constitutively active mouse ACVR1B (ACVR1B$^{T206D}$), a light-oxygen-voltage (LOV) domain, and a C-terminal HA tag (Table EV1). Blue light-dependent dimerization of Opto-receptors was performed as previously described (Ramachandran *et al*, 2018). In brief, NIH-3T3 cells were transfected with a total of 2 μg of plasmid DNA that included 25 ng of FLAG-SMAD1 (Lechleider *et al*, 2001), alone or in combination with varying amounts of either Opto-ACVR1$^{WT}$ or Opto-ACVR1$^{R206H}$. For experiments with Opto-ACVR1B*, cells were transfected as above, but with either 15 ng FLAG-SMAD1 or 5 ng GFP-SMAD3 (Nicolas *et al*, 2004) alone, or in combination with 10 ng Opto-ACVR1B*. Twenty-four hours post-transfection, cells were starved overnight in DMEM with 0.5% FCS. Cells were then left untreated or pre-treated with 1 μM LDN-193189 or 50 μM SB-505124 for 5 min and then exposed to blue light from an LED array for 1 h at 37°C in a humidified incubator. Control cells were kept in the dark by wrapping in aluminum foil and placed in the same incubator

## Antibodies, immunoblotting, and quantification

All primary and secondary antibodies are listed in Table EV2. Western blots using whole cell extracts were performed as previously described (Miller *et al*, 2018). For quantification analysis, see below.

## qPCR

Oligonucleotides used are listed in Table EV1. Total RNA extractions and reverse transcriptions were performed as previously described (Grönroos *et al*, 2012). The cDNA was diluted 10-fold and then used for qPCR. All qPCRs were performed with the PowerUp SYBR Green Master Mix (Thermo Fisher) with 300 nM of each primer and 2 μl of diluted cDNA. Fluorescence acquisition was performed on a 7500 FAST machine (Thermo Fisher). Quantification of relative gene expression was performed using the comparative Ct method with target gene expression normalized to *GAPDH*.

## Confocal microscopy

For live cell confocal imaging, parental, ACVR2A/B dKO, HOM1, and HOM1 ACVR2A/B dKO cells were grown in phenol-free DMEM (Thermo Fisher Scientific) supplemented with 10% FCS, 1 mM sodium pyruvate, and 1% Pen/Strep. Cells were plated on 35 mm High Precision 1.5 MatTek glass-bottom dishes (MatTek, Ashland,

MA, USA) coated with Poly-L-Lysine, or laminin for DIPG cells, and starved overnight in phenol-free DMEM with 0.5% FCS or 3–4 h in tumor stem cell medium without growth factors for DIPG cells. On the day of the experiment, cells were stained for 30 min with 5 µl/ml of CellBrite Green cytoplasmic membrane dye (Biotium) in phenol-free DMEM and then washed according to the manufacturers' instructions.

Imaging was performed on a Zeiss Invert 880 NLO confocal microscope and ZEN platform, with a C-Apochromat 63×/1.2 water objective. His-Activin A Atto647N at 40 ng/ml in phenol-free DMEM was imaged at the start of every Number & Brightness experiment, for calibration of the brightness value for dimeric His-Activin A. For cell imaging, 40 ng/ml of His-Activin A-Atto647N was added to the cells 15 min before the start of imaging. During imaging, cells were kept at 37°C. The green membrane dye was used to select a region of the apical or lateral cell membrane. Images were acquired in raster scan mode, with a frame size of 256×256, pixel size of 0.05 µm and pixel dwell time of 33 µs (scan time/frame of 5.03 s). The 633 nm laser was set at 3%, and a time series of 100 images per ROI were collected in photon-counting mode (Digman *et al*, 2008; Digman *et al*, 2013). The image stacks were transferred to the simFCS software (Laboratory for Fluorescence Dynamics, Irvine, CA) for N&B analysis (Digman *et al*, 2008; Digman *et al*, 2013).

### Preparation of planar lipid bilayers containing His-Activin A and NIP-based antigens

Planar lipid bilayers (PLBs) were prepared by fusing small unilamellar lipid vesicles (SUVs) with clean glass coverslips essentially as described (Tolar *et al*, 2009). Briefly, 1,2-dioleoyl-sn-glycero-3-phosphocholine (DOPC) and 1,2-dioleoyl-sn-Glycero-3-{[N(5-amino-1-carboxypentyl) imino diacetic acid]succinyl} (nickel salt) (DOGS-NiNTA, both from Avanti) were combined in a 10:1 molar ratio in chloroform and dried to a film with a gentle stream of argon and then further under vacuum for 1 h. The lipid film was then resuspended in PBS to 5 mM, vortexed until clear, and then bath sonicated for 30 min to produce SUVs, which were clarified by ultracentrifugation. Glass coverslips (Menzel-Gläser) were cleaned with fresh piranha solution (98% $H_2SO_4$ and 30% $H_2O_2$ in a 2:1 ratio) for 30 min, rinsed thoroughly with Milli-Q $H_2O$, dried with argon, and glued to the bottom of Labtek 8-well imaging chambers (Fisher Scientific). PLBs were prepared from a 0.5 mM solution of SUVs in PBS, and excess lipids were washed away with PBS. PLBs were blocked for 30 min with 0.05% BSA in PBS (w/v), incubated with His-tagged Activin A-Atto647N, His-tagged Activin A-CF640R, or His-tagged 4-hydroxy-5-iodo-3-nitrophenyl acetyl (NIP)$_1$-H12-Hylight647 for 15 min, and excess was washed away with PBS. For imaging, PBS was replaced by phenol-red free DMEM and approximately 250,000 HOM1 cells in phenol-red free DMEM were added to the bilayer and allowed to attach for a few minutes. Time-lapse TIRF imaging was performed to visualize clustering over time.

### TIRF microscopy

TIRF imaging was performed on a Nikon Eclipse Ti microscope with an ORCA-Flash 4.0 V3 digital complementary metal-oxide semiconductor (CMOS) camera (Hamamatsu Photonics) and 100x/1.49 Apo TIRF objective (Nikon), controlled through Metamorph

software (Molecular Devices). Illumination was provided by a 637 nm laser (Cairn) through an iLas2 Targeted Laser Illuminator (Gataca Systems) which produces a 360° spinning beam with adjustable TIRF illumination angle. Automated time-lapse imaging of 3–4 areas of the bilayers containing His-Activin A-Atto647N, or 1–2 areas containing (NIP)$_1$-H12-Hylight647, with HOM1 clones was performed every 40 s, with 100 ms exposure, for 20 min. For Activin A-CF640R with HOM1 clones, time-lapse imaging was performed every 30 s, with 100 ms exposure, for 7.5 min. Acquired time-lapse images were processed into movies at 10 frames per second using FIJI, which was also used for image processing.

### Quantification and statistical analysis

All experiments have been performed with a minimum of two independent biological replicates. Where indicated, data from a representative experiment are presented.

Western blot quantification was performed using the FIJI/ImageJ software; background-subtracted densitometry measurements for pSMAD1/5 were normalized to the loading control (actin or tubulin) and are shown as fold change relative to the indicated condition, unless otherwise specified in the figure legends. Where appropriate, normalized pSMAD1/5 measurements for the different cell lines were used as input into ICEKAT (Olp *et al*, 2020), and initial rates were calculated using default parameters.

Number and Brightness (N&B) analysis provides a single-pixel resolution map of the number (N) and Brightness (B, i.e., oligomerization state) of diffusing particles within the focal spot, by calculating the average (k) and variance (σ) of the intensity distribution at each pixel according to the following formulas (for a detector in photon-counting mode (Digman *et al*, 2008)):

$$B = \frac{\sigma^2}{\langle k \rangle}$$

$$N = \frac{\langle k \rangle^2}{\sigma^2}$$

The value B is related to the molecular brightness ε, defined as counts per second per molecule, and reflects the clustering of fluorophore containing molecules. N is directly proportional to the number of particles *n*. A map of the parameters B and N is used for identifying the regions of an image, i.e., regions of a cell, where protein clustering is occurring. When a given pixel contains a mixture of species with different levels of brightness, the N&B method averages the brightness. Thus, the N&B method cannot accurately measure the size of aggregates but reports on their presence. Photobleaching leads to a decrease in mean intensity over time and can, thus, confound the final results (Trullo *et al*, 2013). To overcome this, we have used a very photostable dye, Atto-647N, which is known to have minimal photobleaching. Images with obvious acquisition artefacts were removed. A high pass filter algorithm (detrend filter) was applied to the time sequence at each pixel of the stack, when mean intensity fluctuations were > 10%, to correct for sample movement and photobleaching during image acquisition (Ossato *et al*, 2010; Trullo *et al*, 2013). The N&B analysis was performed in simFCS (Laboratory for Fluorescence Dynamics).

Fluorescent clusters were detected in areas of the bilayer underneath or outside of cells by a spot detection algorithm using MATLAB similarly as previously described (Roper *et al*, 2019). Briefly, images were flatfield-corrected for illumination and background-subtracted. Spots were then enhanced by convolution with a 325-nm-sized Gaussian filter followed by subtraction of an averaging filter of the same size. The enhanced images were thresholded using an empirically determined intensity value that was kept constant for each experiment. Thresholded areas were watershed-split and used as masks to identify individual clusters and quantify their number and fluorescence intensity in the original corrected images.

Statistical analysis was performed in Prism 8 (GraphPad). At least three independent experiments were performed for statistical analysis. Normalized densitometry measurements from Western blots were log-transformed for analysis by parametric tests. For qPCR data, the $\Delta\Delta$CT value was used in the statistical analysis, and the normalized gene expression data were plotted as individual data points for each biological replicate, with height of the bar representing the mean, and error bars the standard error of the mean (SEM). For pairwise comparison between two groups, paired Student's *t*-test was used, with correction for multiple testing. For comparisons between more than two groups, analysis of variance (ANOVA) was performed, followed by Dunnett's multiple comparison test (for comparison of a number of treatments with a single control) or Tukey's multiple comparison test (for pairwise comparison between all samples). Quantifications of Western blots with 2–3 replicates per condition are presented as individual data points with height of the bar representing the mean, and error bars the standard error of the mean (SEM). For comparison of clustering over time, simple linear regression was performed on mean cluster intensity/cell on Activin and NIP bilayers, and mean cluster intensity on Activin bilayers without cells (corresponding to areas of approximate cell size). Slopes were tested for a non-zero slope, and significant differences between slopes were determined using *F*-test. Outliers were included in statistical analysis. A $P < 0.05$ was considered statistically significant.

# Data availability

This study does not include data amenable for deposition in external repositories.

**Expanded View** for this article is available online.

## Acknowledgements
We thank Paul Yu for LDN-193189 and Bob Lechleider for the FLAG-SMAD1 expression plasmid. We thank Alexandra Stanley for help with the MEFs from *Acvr1*[+/+] and *Acvr1*[+/R206H] mice, Michelle Digman and Enrico Gratton for advice concerning the Number and Brightness analysis, Gavin Kelly and Stuart Horswell for statistics advice, and Louise Richardson for the synopsis image. We are very grateful to the Francis Crick Institute Light Microscopy, particularly Rocco D'Antuono, to the Flow Cytometry Facility, and to the Genomics Equipment Park. We thank all the members of the Hill laboratory and Davide Coda for discussion and very useful comments on the manuscript. For the purpose of Open Access, the corresponding author has applied a CC BY public copyright license to any Author Accepted Manuscript version arising from this submission. This work was supported by the Francis Crick Institute which receives its core funding from Cancer Research UK (FC001095 and FC001185), the UK Medical Research Council (FC001095 and FC001185) and the Wellcome Trust (FC001095 and FC001185), and by Brain Research UK.

## Conflict of interest
The authors declare that they have no conflict of interest.

## Author contributions
AR and CSH conceived and designed the study. AR and MM planned and performed the experiments, and analyzed the data with technical help from LW, DM, PT, and IG. BKB, DMC, EMS, CJ, and MH produced and provided resources. MM performed all the statistical analyses. AR and CSH wrote the manuscript with help from MM. CSH provided supervision and CSH, PT and CJ provided funding for the study.

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
