## [Review Process File · The EMBO Journal]

Pathogenic ACVR1^{R206H} activation by Activin A-induced receptor clustering and autophosphorylation

Anassuya Ramachandran, Merima Mehic, Laabiah Wasim, Dessislava Malinova, Ilaria Gori, Beata Blaszczyk, Diana Carvalho, Eileen Shore, Chris Jones, Marko Hyvonen, Pavel Tolar, and Caroline Hill
DOI: [10.15252/embj.2020106317](https://doi.org/10.15252/embj.2020106317)

Corresponding authors: Caroline Hill (caroline.hill@crick.ac.uk) , Anassuya Ramachandran (anassuya.ramachandran@auckland.ac.nz)

Review Timeline:

Submission Date:	22nd Jul 20
Editorial Decision:	31st Aug 20
Revision Received:	1st Dec 20
Editorial Decision:	1st Dec 20
Revision Received:	21st Jan 21
Editorial Decision:	18th Mar 21
Revision Received:	25th Mar 21
Accepted:	26th Mar 21

Editor: Daniel Klimmeck

Transaction Report:

Dear Dr Hill,

Thank you for the submission of your manuscript (EMBOJ-2020-106317) to The EMBO Journal. Please accept my apologies for the unusual delay with the peer-review of your manuscript due to delayed referee input at this time of the year and detailed discussions in the team. We have sent your manuscript to three reviewers for evaluation and now received reports from all of them, which I copy below. I am afraid that in light of their comments we decided that we cannot offer publication in The EMBO Journal.

As you will see, the referees appreciate the potential interest of your results for the field. However they also raise major concerns with the analysis that I am afraid preclude publication here. In more detail, referee #1 points to major caveats regarding data inconsistencies and not sufficiently supported claims on a kinase activity-independent function of structural receptor clustering in ACVR signaling (ref#1 pts. 5-7). Further, this reviewer is concerned about the pathophysiological relevance of the current ACVR1R206H related findings (ref#1, pts.1, 8). Referee #3 agrees in that the disease relevance of your results is only prematurely explored (Ref#3, standfirst; pt.5). Referee #2 states that the advance of your results as well as the extent of downstream signaling analysis are too limited (ref#2 pts.1,3) and that more experiments are required to conclusively support the model proposed.

Given these negative opinions from good experts on the field, and considering that the journal only offers one concise round of major revisions, I am afraid we have concluded that we cannot offer to publish your study in The EMBO Journal.

Please note that we would per se be able to re-assess a substantially reworked and amended version of the manuscript, but considering the major revision of the study apparently required, as well as the unclear outcome of such experiments, re-evaluation would be entirely open at this point, and you might thus be better advised to seek an alternative venue.

Thank you in any case for the opportunity to consider this manuscript. I regret we cannot be more positive on this occasion, but we hope nevertheless that you will find our referees' comments helpful. I again apologise for the unusual protraction.

Kind regards,

Daniel Klimmeck

Daniel Klimmeck, PhD
Editor
The EMBO Journal

Referee #1:

In this manuscript, Ramachandran et al. identified a new mechanism by which ACVR1R206H promotes TGF β signaling and the pathogenesis of fibrodysplasia ossificans progressiva (FOP) and diffuse intrinsic pontine glioma (DIPG). ACVR1 is a Type 1 TGF β family receptor that is involved in a wide variety of biological processes, including bone, heart, and development regulation. While several ACVR1 mutations have been identified, the majority of DIP and FOP patients exhibit a single amino acid change, Arg206His. Previous studies have shown that this point mutation results in enhanced Smad 1/5-induced signaling. However, the underlying mechanisms are not well understood. To address this, the authors introduced a point mutation in the endogenous ACVR1 locus of HEK23T cells and demonstrated that ACVR1R206H exhibited an enhanced response to Activin A compared to ACVR1WT. Mechanistically, the authors show that Activin A promotes clustering of ACVR1R206H with ACVR2A/B at the cell surface and this clustering is sufficient for ACVR1R206H autoactivation and Smad 1/5 phosphorylation.

The novelty of this study lies in the findings that ACVR1R206H activation does not require upstream kinases, but is largely activated by Activin A-dependent receptor clustering. Prior to this study, the accepted model of TGF β signaling centered on constitutively active T β RII phosphorylating T β RI to activate Smad 1/5. The present study introduces a new paradigm to TGF β signaling, especially in the pathogenesis of FOP and DIP. Moreover, this study is one of the first to show that T β RI activation can occur independently of T β RII kinase activity. The overall conclusion appears sound and the identification of a novel mechanism for FOP and DIPG is very promising. However, a number of additional experiments need to be performed to fully support their overall findings, and the translational application of this study is unclear.

Major Comments:

1. A number of experiments were performed using H293T cells. While these cells are sufficient for preliminary studies, more physiological relevant cell lines should be used.
2. For Figure 2, the authors should provide direct evidence that ACVR1R206H is not internalized following Activin A stimulation and is localized at the cell surface through immunofluorescence studies or biotinylating/streptavidin labeling. Moreover, Smad 1/5 nuclear accumulation should also be investigated.
3. The conclusion that ACVR1R206H signals independently of ACVR1B/C at higher concentrations of Activin A are unsupported. Slight impairment of Smad 1/5 activation at high concentrations of Activin A can largely be explained by the fact that higher SB-431542 concentration that was used was not sufficient enough to fully inhibit receptor activity. Moreover, this conclusion is also not supported by Figure 3B as dKO of ACVR1B/C significantly impaired Smad 1/5 activation in HOM1 cells (as indicated by the representative western blot image). More direct evidence is needed to show that ACVR1R206H signals independently of ACVR1 B/C at higher concentrations of Activin A.
4. The authors argue that the kinase activity is not needed for ACVR1R206H - mediated induction of Smad 1/5. However, in Figure 4C, overexpression of ACVR2AKR failed to rescue Smad 1/5 activation compared to ACVR2AWT cells, suggesting that the kinase domain of type II receptors is critical for ACVR1R206H function. The authors should address this.
5. In Figure 4B, it is unclear why the authors overexpress pFlag-Smad1 for their optogenetic studies. The western blot clearly shows that endogenous pSmad 1/5 can be detected. While there is a large change in pFlag-Smad1 upon light-induced clustering of ACVR1R206H, there is very little to no change in endogenous pSmad 1/5 between ACVR1WT and ACVR1R206H groups, suggesting that

Smad 1 activation by ACVR1R206H-mediated clustering may be an artifact. The authors should address this discrepancy.

6. While TIRF microscopy data is intriguing and dramatic, it does not directly demonstrate that Activin A promotes ACVR1R206H and ACVR2A/B clustering, rather it only demonstrates that the lipid bilayer conformation is altered. More direct evidence demonstrating that Activin A selectively promotes ACVR1R206H and ACVR2A/B clustering is needed.

7. The authors claim that ACVRIIA/B promotes cluster formation structurally rather than through its kinase activity as double knockout of ACVRII A/B resulted in a significant reduction in tetramers and oligomers. This is not surprising as one would expect higher order structures to be altered if total levels of a receptor are depleted. To fully demonstrate that ACVRII A/B structure but not enzymatic activity is needed for cluster formation, the authors should treat cells with an ACVRII A/B specific inhibitor or use a KD (kinase dead) ACVRII A/B and reperform Figure 6.

8. While the signaling data is strong, there is a lack of clinical relevance. The authors should demonstrate that receptor clustering occurs in FOP mouse models or in human samples. Moreover, the authors should show that FOP pathogenesis and receptor clustering can be attenuated using an ACVR1 inhibitor. These studies should also be performed *in vitro*.

9. The conclusion that ACVR1R206H undergoes autoactivation is unsupported in this manuscript. To fully demonstrate this, the authors should perform proteomic analysis of ACVR1R206H in the presence or absence of ACVRII A/B.

Minor Comments:

1. In Figure 1A, the authors claim that ACVR1R206H specifically promotes Activin A-mediated Smad 1/5 activation and not BMP-mediated Smad 2/3 activation. However, based on the western blot image and fold change, ACVR1R206H also enhances BMP-mediated Smad2/3 activation. The authors should address this.

2. It is unclear if high concentrations of Activin A were used in Figure 3B.

3. Longer exposure for Flag should be shown to demonstrate that at ACVR2AWT and ACVR2AKR are being expressed at 20 ng and 200 ng.

4. The authors should perform confocal and live cell imaging to supplement their findings from Figure 6A.

5. The authors demonstrate that Activin A stimulation induces receptor clustering of Type I and Type II receptors to promote Smad 1/5/8 activation but they did not address whether Activin A promotes the overall stability of this complex at the cell surface.

6. Alterations in Smad 1/5 signaling should be supplemented with the expression of Smad 1/5-target genes.

Referee #2:

In this manuscript the authors explore Activin signaling through ACVR1, also known as ALK2. When ACVR1 was first cloned it was identified as an activin receptor, and then subsequently a BMP receptor. Since then it has become clear that activin can engage ACVR1 to activate Smad1 and Smad5 in certain contexts, as well as ACTR1B/C more generally to induce Smad2/3 activation (see for example, Olson et al., 2020. *Biomolecules* 2020, 10(4), 519). This dual specificity of activins is dependent on context, since Activins can also antagonize BMP signalling in certain systems. A specific mutation in ACVR1 (R206H) causing fibrodysplasia ossificans progressiva (FOP) was shown in 2015 to promote Activin-dependent signaling through Smad1/5. In these models Activin maintains a TGFbeta-like pathway and forms a non-signaling complex with ACVR1 presumably to

buffer Smad1 signaling. In contrast, the R206H mutant leads to aberrant activin-dependent activation of Smad1/5. So the prior literature suggests that in certain contexts, particularly with respect to the R206H mutant, Activin A can induce BMP-Smad signaling and disease.

Here the authors explore ACVR1 signaling downstream of Activin and report that ACVR1 mediates Activin-dependent Smad1/5 activation through a canonical type II receptor-dependent pathway that also employs ACVR1B/C. This is analogous to previous studies showing that in endothelial cells TBR1 mediates recruitment and optimal activation of the ACVR1-like type I receptor, ALK1, to mediate TGFbeta-dependent activation of Smad1/5 in endothelial cells (Goumans et al., Mol. Cell, 2003; EMBO J., 2002). The current manuscript attempts to extend this prior work to an activin pathway, and additionally proposes that as in prior publications, the R206H mutant converts ACVR1 into an activin-dependent BMP receptor. Here the authors add to this work by proposing the mutation might induce receptor clustering and promiscuous kinase activity, which supports previously published structural studies of ACVR1. Overall, this paper explores how the R206H mutation in ACVR1 might function at the molecular level, but very little of the conclusions are all that novel, and rather extend prior observations to this particular model. As noted below, the studies to try and demonstrate an ACVR2:ACVR1B/C:ACVR1 signaling complex in wild type cells are highly problematic. In sum, I'm puzzled by what "new paradigm" is revealed in this manuscript. Furthermore, the paper relies on a single cell line model, which is highly problematic, given the important role of cell context of TGFbeta family dual pathway signaling highlighted above.

Specific comments:

1) The biggest issue with this manuscript is that the entire model for wild type signaling is based on inferring a molecular model of receptor interactions by only measuring Smad activation. Critical to support the authors' model is showing a ligand receptor complex comprised of ACVR2:ACVR1B/C:ACVR1, showing that within this complex that ACVR2 phosphorylates ACVR1B/C, and then showing that ACVR1B/C phosphorylates ACVR1. Relying entirely on Smad activation is highly problematic and the results can be interpreted in many different ways. For example, activin could signal through its canonical receptor pathway to modulate an autocrine BMP response. Evidence in the manuscript certainly points to autocrine BMP signaling present in this cell type.

2) In Figure 1 the authors make mutant HEK293 cell lines where they introduce the R206H mutant and KO ACVR1. The results are largely consistent with the known literature. What's puzzling is that while mutant ACVR1 promotes Activin A signaling it does not affect Activin B. At the molecular level it's difficult to reconcile this with the future conclusions about ligand induced ACVR1B/C receptor clustering. Switching to analysis of BMP2 in C and D is puzzling since BMP2 employs additional type I receptors. BMP4:7 should be used for comparative analyses.

3) Figure 2 provides a bit more information on Activin A temporal induction of Smad1/5 and adds to the previous studies of Hatsell, Hino and more recently Olsen. The authors continue comparison of activin A with BMP2 in temporal kinetics. This is problematic as BMP2 signaling is not shown to be dependent on ACVR1 in this model. They need to use BMP4:7 dimers, or BMP6 or BMP7, which require ACVR1, particularly as they are using these data to conclude that ACVR1 and the R206H variant do not show substantive differences in trafficking. They then explore the I-SMAD negative feedback loop and propose that Smad2/3-mediated induction of I-SMADs regulate ACVR1. However, BMPs also induce I-SMADs, which was not explored in this model and they did not assess whether I-SMAD KOs affected extended signaling by the mutant receptor. In particular, the time course of Activin A signaling through mutant receptors in EV3A is extended compared to WT, and so could be consistent with R206H promoting cell surface levels. This highlights a challenge with

this part of the manuscript in that it only employs indirect measures of receptor activity to make strong conclusions about WT and R206H having "broadly similar intracellular signaling dynamics and recycling". Directly measuring cell surface and endosomal receptor is required.

4) In Figure 3 the authors present experiments that suggest ACVR1B/C is important for Activin-dependent ACVR1 signaling to BMP Smads. The problem with these experiments is that the cell model displays marginal Activin-dependent Smad1 activation (only about 3 fold at best and about ~10-fold less than the BMP response, see Fig. 1), so the pathway is not active in these cells, which makes interpretation of the results challenging, especially since KO lines are essentially clonal derivatives of the parental. Thus, even slight clonal variation in responsiveness could account for the observations on WT ACVR1 cells in panel B. Of note, recent studies (Aykul et al., eLife, 2020), also in HEK293 cells, show no evidence of activin-dependent activation of Smad1 either by blotting for P-Smad1 or assessing a BMP reporter construct. It's unclear why the current studies show these trace BMP activation events.

Note that panel A is very noisy and certainly shows evidence of SB421542 inhibition of the R206H mutant. In panel B BMP4:7 should be used for the comparison. The data suggesting ACVR1B/C is not required for R206H activity is convincing.

5). The authors also attempt to show that TGFbeta induces p-Smad1 in MEFs. Unfortunately, the data is even less convincing than the conclusion that Activin A induces P-Smad1. What's the biological significance of such a marginal effect? This experiment is also noisy and the authors should employ BMP4:7 for comparative purposes.

6) Having established that ACVR1B/C are not important for activin-dependent R206H signaling, the authors tackle the role of the upstream type II receptors, ACVR2A/B. They show that type II receptors and their catalytic activity are important for signaling in the wild type scenario, but then show a kinase dead ACVR2A can support R206H signaling. They use optogenetics to induce receptor clusters and show that artificially induced clustering of R206H induces activation. An important observation here that is ignored by the authors is that in the absence of light the R206H mutant also clusters at higher expression levels. So this mutation provides a propensity for type I receptors to cluster and yield promiscuous autophosphorylation that at endogenous levels is promoted by the type II receptor. This supports previous conclusions about how the mutant converts into an Activin->BMP Smad signaling complex as well as structure-based studies.

7). At the top of page 11 the authors make the strange conclusion that since light-induced ACVR1 clusters fail to activate Smad1, then this demonstrates the key requirement for ACVR1B/C. But BMP signaling through ACVR1 does not require ACVR1B/C so there is no reason why ACVR1 in such an artificial system would not activate Smad1 except for the fact no type II kinase is present in these light-induced clusters.

8) In the final experiments the authors go on to hypothesize that in vivo Activin A induces type II-type I complexes and receptor clustering. I found this phrasing rather odd given that this has been amply demonstrated in the prior literature including structural studies of higher order receptor ligand complexes, all of which is confirmed here, so I don't understand the value of these experiments in the context of the current paper.

9). Throughout the manuscript there are numerous experiments that lead to important conclusions that appear to have only been done once. Eg Fig. 1A, EV3A et al. I also find the use of mean +/- SEM for experiments done only twice to be inappropriate.

Referee #3:

The manuscript by Ramachandran et al. investigates the mechanisms through which the wildtype Activin Type 1 receptor (ACVR1WT) and a "gain-of-function" mutant form (ACVR1R206H) are activated in response to Activin stimulation. Through a series of biochemical experiments, the authors elucidate two distinct mechanisms governing the activation of these receptors. In the case of ACVR1WT, the authors determine that Activin A can bind to ACVR2A/B and ACVR1B/C, which in turn phosphorylates and activates ACVR1WT. A similar mechanism may also activate ACVR1R206H at low activin concentrations. However, at high activin concentrations the authors demonstrate that ACVR1R206H can form direct complexes with ACVR2A/B. In this latter scenario, the catalytic domain of the type II activin receptors (ACVR2A/B) are dispensable, contributing a scaffolding function that enhances signaling via ACVR1R206H.

The ACVR1R206H receptor is implicated in two diseases, which include Fibrodysplasia ossificans progressiva (FOP) and diffuse intrinsic pontine glioma (DIPG). Thus, considerable effort has been invested into determining how the ACVR1R206H receptor is constitutively activated. The current study uses extensive biochemical assays and sophisticated approaches to generate data that supports the novel mechanisms proposed in the paper. This paper will be of interest to researchers in this field. One aspect that is missing from the current study is an attempt to assess if the proposed mechanisms elucidated in this paper are operative in disease models in which this gain of function receptor contributes to the disease (e.g. FOP).

Specific Points:

1) The authors state that HEK293T cells that are homozygous for the ACVR1R206H mutation exhibit dramatically increased pSmad1/5 levels compared to parental HEK293T cells. While this is true for the HOM1 cells in Figure 1, this does not appear to be the case for HOM2 cells (Figure EV1). For example, the pSmad1/5:Actin ratio in parental HEK293T cells is increased 29 fold following Activin A treatment compared to a 253fold increase in HOM1 cells (Figure 1A). In contrast, in Figure EV1C, Activin increases the pSmad1/5:Actin ratio by 13-fold relative to unstimulated HEK293T parental cells and only 10-fold in HOM2 cells. Can the authors comment on this discrepancy?

2) In the discussion, the authors speculate that the ACVR1R206H receptor could undergo differential ubiquitination and lysosomal degradation (reduced relative to wildtype ACVR1), which could lead to prolonged signaling. In light of this, it would be important to show that similar levels of ACVR1WT is expressed in wildtype HEK293T cells when compared to ACVR1R206H in HOM1 or HOM2 cells by immunoblot approaches.

3) In line with the second comment, the authors have created a unique dimeric Activin A ligand that is tagged with a fluorophore (His-Activin A-Atto647N: used in Figure 6 and 7). Could live-imaging approaches be used to trace activin-bound ACVR1WT or ACVR1R206H receptors with co-labelling with an early endosomal marker (EEA1) or a lysosomal marker (LAMP1). This would corroborate the conclusions drawn from Figure 3 (no difference between endosomal signaling between the two receptors) and determine if less ACVR1R206H is degraded.

4) Non-canonical pathways, beyond Smad1/5, have also been suggested to be activated downstream of ACVR1R206H, most notably p38 activation. Have the authors assessed p-p38 and total p38 levels in parental, HET and HOM cells following Activin stimulation?

5) The strength of the paper lies in the rigorous biochemical characterization of these receptor activation mechanisms in a tractable cell model. However, it would strengthen the paper considerably if supporting evidence for this activation model could be obtained using samples in which expression of ACVR1R206H is implicated in the pathology. Indeed, knock in mice harboring the ACVR1R206H allele develop phenotypes reminiscent of those seen in FOP patients. Significant data also indicates that Activin A can contribute to excessive ossification in models of FOP. Could samples (ossified skeletal muscle for example) from the ACVR1R206H knock in model be assessed for complex formation between ACVR2A/B receptors with ACVR1R206H using proximity ligation assays (PLAs). This would depend on antibodies that could detect the type II and type I receptors. One might predict, based on the model, that high PLA signals should be detected between ACVR2A/B and ACVR1R206H. If such samples are impossible to get, could the MEFs be used (described in Figure EV5). PLA assays in +/- ACVR1R206H MEFs would show a low PLA signal between ACVR2A/B and ACVR1R206H under low Activin A conditions but an increased association between the type II receptors and gain of function type I receptor under high activin A concentrations, if the model is correct. A low PLA signal would be expected in the wildtype MEFs. These experiments would add substantially to the paper.

** As a service to authors, EMBO Press provides authors with the possibility to transfer a manuscript that one journal cannot offer to publish to another EMBO publication or the open access journal Life Science Alliance launched in partnership between EMBO Press, Rockefeller University Press and Cold Spring Harbor Laboratory Press. The full manuscript and if applicable, reviewers' reports, are automatically sent to the receiving journal to allow for fast handling and a prompt decision on your manuscript. For more details of this service, and to transfer your manuscript please click on Link Not Available. **

Dear Dr Klimmeck,

Many thanks for your e-mail regarding the reviews of our manuscript (EMBOJ-2020-106317). We were very pleased to see that the reviewers appreciated the potential interest of our results for the field. Reviewer 1 thought that the overall conclusions appear sound and that the identification of a novel mechanism for FOP and DIPG is very promising. Reviewer 3 appreciated that our study uses extensive and rigorous biochemical assays and sophisticated approaches to generate data that supports the novel mechanisms proposed in the paper.

You indicated in your decision letter that you would be able to re-assess a substantially reworked and amended version of the manuscript. We have been through all the reviewers' comments and realised that we could readily address all of them with additional experiments or with textual clarifications, where the comments arose from misunderstandings, which was particularly the case with Reviewer 1's comments.

I have waited to write back to you until I knew for certain that we could actually perform all the necessary experiments in these difficult restricted COVID times. We have now almost completed them, so I am writing with a point-by-point rebuttal to show you how we can address the reviewers' comments and criticisms. Most importantly, we have been able to increase the clinical relevance of our study with the inclusion of tumour cells from DIPG patients that have either ACVR1WT or ACVR1R206H, obtained from our collaborator, Prof Chris Jones (ICR).

Once you have had a chance to look at the rebuttal, I would like to know whether you would consider the revised manuscript for publication in EMBO J. We would be ready to submit a substantially reworked manuscript in a couple of weeks' time.

I look forward to hearing from you soon.

Best wishes

Caroline Hill

Dear Caroline,

Thank you for contacting me regarding our decision and for your patience with my feedback, which got delayed due to internal discussions in the team regarding your point-by-point response.

We realise that you would - judging from the information provided in the rebuttal letter - be potentially able to address the issues raised by the referees in a revised version of the manuscript.

Overall, we would thus invite you to work towards resubmission and re-review. Accordingly, we would - given that you addressed the experimental issues with compelling data - be prepared to ask the referees for further input.

Please note however, that we would need the major concerns of the reviewers regarding conclusive support on the signaling mechanism, as well as the pathophysiological relevance of your findings to be satisfactorily addressed. In that sense, it would be essential to provide a definitive and accurately described dataset in the revised version.

Please contact me if you have any questions, need further input on the referee comments or if you consider engaging in a compelling revision, in which case we would not close the file.

Kind regards,

Daniel

Daniel Klimmeck, PhD
Senior Editor
The EMBO Journal

IMPORTANT: When you send the revision we will require
- a point-by-point response to the referees' comments, with a detailed description of the changes made (as a word file).

- a word file of the manuscript text.
 - individual production quality figure files (one file per figure)
 - a complete author checklist, which you can download from our author guidelines (<https://www.embopress.org/page/journal/14602075/authorguide>).
 - Expanded View files (replacing Supplementary Information)
- Please see out instructions to authors
<https://www.embopress.org/page/journal/14602075/authorguide#expandedview>

The revision must be submitted online within 90 days; please click on the link below to submit the revision online before 1st Mar 2021.

Link Not Available

Rebuttal to reviewers' comments: EMBO J-2020-106317

Referee #1:

In this manuscript, Ramachandran et al. identified a new mechanism by which ACVR1R206H promotes TGF β signaling and the pathogenesis of fibrodysplasia ossificans progressiva (FOP) and diffuse intrinsic pontine glioma (DIPG). ACVR1 is a Type 1 TGF β family receptor that is involved in a wide variety of biological processes, including bone, heart, and development regulation. While several ACVR1 mutations have been identified, the majority of DIP and FOP patients exhibit a single amino acid change, Arg206His. Previous studies have shown that this point mutation results in enhanced Smad 1/5-induced signaling. However, the underlying mechanisms are not well understood. To address this, the authors introduced a point mutation in the endogenous ACVR1 locus of HEK293T cells and demonstrated that ACVR1R206H exhibited an enhanced response to Activin A compared to ACVR1WT. Mechanistically, the authors show that Activin A promotes clustering of ACVR1R206H with ACVR2A/B at the cell surface and this clustering is sufficient for ACVR1R206H autoactivation and Smad 1/5 phosphorylation.

The novelty of this study lies in the findings that ACVR1R206H activation does not require upstream kinases, but is largely activated by Activin A-dependent receptor clustering. Prior to this study, the accepted model of TGF β signaling centered on constitutively active T β RII phosphorylating T β RI to activate Smad 1/5. The present study introduces a new paradigm to TGF β signaling, especially in the pathogenesis of FOP and DIPG. Moreover, this study is one of the first to show that T β RI activation can occur independently of T β RII kinase activity. The overall conclusion appears sound and the identification of a novel mechanism for FOP and DIPG is very promising. However, a number of additional experiments need to be performed to fully support their overall findings, and the translational application of this study is unclear.

We thank the reviewer for appreciating the new mechanism we propose in this paper and for recognising its significance. For clarification, this manuscript primarily focuses on the Activin receptors (ACVR2A/B, ACVR1B/C) and on ACVR1, although we do touch on the mechanism of TGF- β -mediated activation of ACVR1^{R206H} (**Appendix Figure S2**) and generally believe it to be similar to Activin in this context.

Major Comments:

1. A number of experiments were performed using H293T cells. While these cells are sufficient for preliminary studies, more physiological relevant cell lines should be used.

HEK293T cells were chosen as a model as they are very amenable to genetic modification and responsive to Activin signalling. To strengthen the mechanistic implications of this study, and to investigate the disease relevance, we have now additionally used tumour cells from DIPG patients that have either ACVR1^{WT} or ACVR1^{R206H}, obtained from our collaborator, Prof Chris Jones (ICR) to perform key experiments (**new Figure 3C, new Figure 6i, new Figure EV4B-F**). We show that in cells expressing ACVR1^{R206H}, Activin A-induced pSMAD1/5 is independent of ACVR1B/C as we see in HEK293T cells. Most importantly, we demonstrate Activin A-dependent receptor clustering in these patient cell lines, which is stronger for the line containing ACVR1^{R206H}, and in both cell lines is dependent on the type II receptors ACVR2A/B.

2. For Figure 2, the authors should provide direct evidence that ACVR1R206H is not internalized following Activin A stimulation and is localized at the cell surface through immunofluorescence studies or biotinylating/streptavidin labeling. Moreover, Smad 1/5 nuclear accumulation should also be investigated.

Based on our previous work (Miller et al, 2018; 2019, PMIDs 30428352 and 31217285 respectively) and the experiments shown in **Figure EV3**, we believe that both ACVR1^{WT} and ACVR1^{R206H} are internalised upon Activin A binding. Indeed, from the Follistatin chase experiments, we conclude that the internalisation/recycling rates for both ACVR1^{WT} and ACVR1^{R206H} are equivalent (**Figure EV3**).

Despite the many years for which the receptors have been known and the vast body of literature on TGF- β family signalling, there are very few antibodies that reproducibly and reliably detect endogenous receptors and there are none that can detect endogenous ACVR1 in any cell type. I should stress that we have tested many antibodies over the last 20 years. Therefore, at present, we cannot study the dynamics of endogenous ACVR1, either by immunofluorescence or biotinylation/streptavidin labelling. While transgenic receptors can be monitored, overexpression greatly alters the physiological balance of receptors in different compartments and this would confound the results.

We have attempted to use our fluorescently-labelled Activin A to label endogenous receptors, but unfortunately, this was not successful as the signal-to-noise ratio was just too low, even using a super resolution microscope.

The reviewer suggests looking at SMAD1/5 nuclear accumulation. We have chosen not to do this as SMAD1 and SMAD5 tend to be quite nuclear even in the absence of signal; they are imported slightly faster than they are exported. Thus, the differential between untreated and treated cells is not very striking. See for example Xiao et al 2001, PMID 11509558, Figure 2 and Figure 4. We find that using Western blotting for pSMAD1/5 is much more quantitative, as we are averaging thousands of cells, and the range of the assay is very high.

3. The conclusion that ACVR1R206H signals independently of ACVR1B/C at higher concentrations of Activin A are unsupported. Slight impairment of Smad 1/5 activation at high concentrations of Activin A can largely be explained by the fact that higher SB-431542 concentration that was used was not sufficient enough to fully inhibit receptor activity. Moreover, this conclusion is also not supported by Figure 3B as dKO of ACVR1B/C significantly impaired Smad 1/5 activation in HOM1 cells (as indicated by the representative western blot image). More direct evidence is needed to show that ACVR1R206H signals independently of ACVR1 B/C at higher concentrations of Activin A.

We thank the reviewer for raising a point that was perhaps not sufficiently clear in the manuscript. First, to address the technical issue raised, the concentration of SB-431542 used in these experiments (10 μ M) is not limiting. It completely inhibits SMAD2 phosphorylation mediated by Activin A in parental, HET and HOM1 cells and so is sufficient to inhibit all kinase activity of ACVR1B/C. This is now made clear in the text (**page 9, paragraph 2**). Higher concentrations will likely lead to off-target effects on other kinases, making the interpretation of results very difficult.

The results in **Figure 3** demonstrate that Activin A achieves phosphorylation of SMAD1/5 by two mechanisms depending on dose of Activin A and status of ACVR1. In parental cells, at both doses of Activin A, low or high, Activin A-induced pSMAD1/5 is inhibited by SB-431542 or by ACVR1B/C dKO, showing that it is entirely dependent on ACVR1B/C. In ACVR1^{R206H} cells treated with a low dose of Activin A, SB-431542 also substantially inhibits induction of pSMAD1/5. Thus, at low dose, Activin A-induced pSMAD1/5 in ACVR1^{R206H} cells is dependent on ACVR1B/C. However, at high doses of Activin A in ACVR1^{R206H} cells, significant pSMAD1/5 persisted even after treatment with SB-431542 or in ACVR1B/C dKO cells. This is clear in both the representative westerns and in the quantitations (**Figure 3A, B and D**). This indicates that at high doses, Activin A *additionally* induces pSMAD1/5 via ACVR1^{R206H}, independently of ACVR1B/C. The ACVR1B/C dKO cells are an excellent control in these experiments, as this approach overcomes any limitation of the use of small molecule inhibitors. We show later in the paper that the underlying mechanism involves the high dose of Activin A clustering ACVR1^{R206H} to activate it, which is independent of the kinase activity of either ACVR1B/C or the type II receptors. Moreover, consistent with our findings in the knockin HEK293T cells, we have now shown that in the ACVR1^{R206H} DIPG cell line HSJD-DIPG-007, SB-431542 has very little effect on Activin A-induced pSMAD1/5, although it completely inhibits Activin A-induced pSMAD2 (**new Figure 3C**). This finding demonstrates that this mechanism also operates in the disease context.

Further support for ACVR1B/C-independent activation of ACVR1^{R206H} is evident in the optogenetic experiments in **Figure 5** where light-dependent clustering of ACVR1^{R206H} alone is sufficient for pSMAD1/5, without a need for ACVR1B/C or a type II receptor. Thus, three independent lines of evidence demonstrate that ACVR1^{R206H} can be activated independently of ACVR1B/C. We have improved the text in the results in the manuscript (**pages 9, 10 and 11**) to make this clearer.

4. The authors argue that the kinase activity is not needed for ACVR1R206H-mediated induction of Smad 1/5. However, in Figure 4C, overexpression of ACVR2AKR failed to rescue Smad 1/5 activation compared to ACVR2AWT cells, suggesting that the kinase domain of type II receptors is critical for ACVR1R206H function. The authors should address this.

In **Figure 4A**, we demonstrate that in both parental and HOM1 cells, loss of ACVR2A/B leads to loss of Activin A-dependent pSMAD1/5. In **Figure 4B**, we demonstrate that in parental cells, only ACVR2A^{WT} could restore pSMAD1/5 in response to Activin A. In **Figure 4C**, we demonstrate that in HOM1 cells, **both** ACVR2A^{WT} **and** ACVR2A kinase dead (ACVR2A^{KR}) restore pSMAD1/5 downstream of Activin A. Thus, it is clear that ACVR2A^{KR} can only complement ACVR1^{R206H} **and not** ACVR1^{WT}. We have improved the text in the results (**pages 10–11**) to make this clearer.

It is true that the level of pSMAD1/5 in transiently-transfected HOM1 ACVR2A/B dKO cells is not equivalent to that in HOM1 cells. However, this is most likely due to the transfection efficiency, the delivery of sufficient ACVR2A to only a proportion of the cells, and the difficulty in getting transfected receptors to phosphorylate endogenous SMADs (please also see response under point 5 below).

5. In Figure 4B, it is unclear why the authors overexpress pFlag-Smad1 for their optogenetic studies. The western blot clearly shows that endogenous pSmad 1/5 can be detected. While there is a large change in pFlag-Smad1 upon light-induced clustering of ACVR1206H, there

is very little to no change in endogenous pSmad 1/5 between ACVR1WT and ACVR1R206H groups, suggesting that Smad 1 activation by ACVR1R206H-mediated clustering may be an artifact. The authors should address this discrepancy.

We thank the reviewer for raising this important point. We used FLAG-SMAD1 in the assays because, in our experience of overexpression of wildtype receptors in many different cell lines, we had previously found that they rarely phosphorylated endogenous SMADs very well (please see response to point 4 above) and were much better assayed by co-transfecting the substrate SMAD, possibly because of relatively low transfection efficiency (for a nice example of this see Figure 7A in Randall et al., 2004, PMID 14729957). The reviewer is correct that some phosphorylation of endogenous SMAD1/5 by Opto receptors is evident in these experiments. However, our ability to detect this is highly dependent on the transfection efficiency of the experiment. We have, therefore, retained FLAG-SMAD1 in these assay as it substantially extends the range and reproducibility of the assay. We have now included a sentence in the results (**page 11, paragraph 2**) to address this point.

6. While TIRF microscopy data is intriguing and dramatic, it does not directly demonstrate that Activin A promotes ACVR1R206H and ACVR2A/B clustering, rather it only demonstrates that the lipid bilayer conformation is altered. More direct evidence demonstrating that Activin A selectively promotes ACVR1R206H and ACVR2A/B clustering is needed.

This is an important point. The TIRF methodology used in this study is powerful and often used to study B- and T-cell receptor clustering (references in our manuscript and further PMID: 20054396, 16200067, 6333027 and 24411234). Lipid bilayers formed in this way diffuse laterally and longitudinally (i.e. X and Y planes), but not vertically (i.e. Z) and so cannot deform. Since the bilayer is mobile, as is the fluorescent Activin A ligand bound to it, any increase in fluorescence intensity is the result of focused clustering in the X/Y plane. We have now clarified the text in results (**page 13, paragraph 3**) to address this point.

As a control, we coupled 4-hydroxy-5-iodo-3-nitrophenyl acetyl (NIP)₁-H12-Hylight647 to the bilayer. (NIP)₁ is a hapten that is recognised by B-cells (PMID: 20620943) and does not bind Activin receptors. Indeed, plating HOM1 cells on (NIP)₁ did not result in any clustering (**Figure 7A**). Furthermore, neither His-Activin A-Atto647N nor His-Activin-A-CF640R clustered on their own in the absence of HOM1 cells (**Figure 7B, C, D and Figure EV5A, C, D**). Thus, the lipid bilayer does not spontaneously cluster. Taken together with the Number and Brightness data in **Figure 6** and the data in **Figure 4**, we conclude that Activin A binding induces higher order clustering of ACVR1^{R206H} which depends on the presence of the type II receptors, though not their kinase activity.

7. The authors claim that ACVRIIA/B promotes cluster formation structurally rather than through its kinase activity as double knockout of ACVRII A/B resulted in a significant reduction in tetramers and oligomers. This is not surprising as one would expect higher order structures to be altered if total levels of a receptor are depleted. To fully demonstrate that ACVRII A/B structure but not enzymatic activity is needed for cluster formation, the authors should treat cells with an ACVRII A/B specific inhibitor or use a KD (kinase dead) ACVRII A/B and reperform Figure 6.

The reviewer makes an important point that knockout of the type II receptors is not sufficient to prove that they are required for the clustering – a specific inhibitor of ACVR2A/B is also

required. Indeed, this is exactly what we have done using ligand traps comprising the extracellular domain of human ACVR2A and ACVR2B fused to the Fc portion of human IgG1 (rhACVR2A/B Fc). These act as competitive inhibitors of ACVR2A/B. We demonstrated that both in parental and HOM1 cells, the rhACVR2A/B Fc ligand traps prevented Activin A-mediated clustering (**Figure 6D and Figure 6G** respectively). We believe that this experiment, taken together with data in **Figure 4 and Figure 7**, demonstrate that Activin A promotes clustering and activation of ACVR1^{R206H} via a structural requirement of ACVR2A/2B for ligand binding.

8. While the signaling data is strong, there is a lack of clinical relevance. The authors should demonstrate that receptor clustering occurs in FOP mouse models or in human samples. Moreover, the authors should show that FOP pathogenesis and receptor clustering can be attenuated using an ACVR1 inhibitor. These studies should be also be performed in vitro.

We were pleased to see that the reviewer finds the signalling data we have generated in this manuscript to be strong. To strengthen the clinical relevance of this work, we have additionally performed the experiments with the type I receptor small molecule inhibitors and the live imaging receptor clustering analyses in DIPG cell lines derived from patients, that have either ACVR1^{WT} or ACVR1^{R206H}, obtained from our collaborator, Prof Chris Jones (ICR) (**new Figure 3C, new Figure 6I and new Figure EV4B-F**). Reassuringly, results from these disease relevant models are in good agreement with our data in HEK293T cells.

9. The conclusion that ACVR1R206H undergoes autoactivation is unsupported in this manuscript. To fully demonstrate this, the authors should perform proteomic analysis of ACVR1R206H in the presence or absence of ACVR2A/B.

The optogenetics experiments provide very strong data to support the conclusion that ACVR1^{R206H} undergoes autoactivation upon clustering. We show clearly in **Figure 5** that Opto-ACVR1^{R206H}, but not Opto-ACVR1^{WT} is activated and can phosphorylate SMAD1 upon blue-light stimulation. The reviewer's suggestion to further confirm receptor activation by proteomics is a good one. However, we have tried unsuccessfully in the past to map the phosphorylation sites in the GS domain of type I receptors, including ACVR1. The problem we encountered was the lack of suitable protease cleavage sites within and around the GS domain. This meant that fragments generated were not suitable for mass spectrometry analysis/mapping of phospho-peptides in the GS domain.

Minor Comments:

1. In Figure 1A, the authors claim that ACVR1R206H specifically promotes Activin A-mediated Smad 1/5 activation and not BMP-mediated Smad 2/3 activation. However, based on the western blot image and fold change, ACVR1R206H also enhances BMP-mediated Smad2/3 activation. The authors should address this.

In our hands, unlike TGF- β s and Activins, BMPs are unable to mediate SMAD2/3 phosphorylation and there is no evidence of this in the data in the original manuscript. However, while performing experiments in the ACVR1^{R206H} DIPG line, HSJD-DIPG-007, we unexpectedly observed pSMAD2 upon treatment with BMP4/7 (**new Figure 3C**). We have previously observed that recombinant ACVR1 is capable of efficiently phosphorylating SMAD2 *in vitro* (Ramachandran et al 2018, PMID 29376829), raising the possibility that ACVR1^{R206H} may confer a neofunction to BMP4/7.

2. It is unclear if high concentrations of Activin A were used in Figure 3B.

We are sorry that this did not come across clearly. In the interests of brevity in the manuscript and to keep figure legends consistent, we have stated in the Methods section under “Ligands, chemicals and cell culture” that “unless otherwise noted, all ligands were used at 20 ng/mL”. However, we have now also included this detail in the figure legend.

3. Longer exposure for Flag should be shown to demonstrate that at ACVR2AWT and ACVR2AKR are being expressed at 20 ng and 200 ng.

Longer exposures unfortunately do not reveal any band at the lower concentrations of transfected receptors in **Figure 4**. Thus, the expression achieved at these levels is below the detection limit of the Western blotting reagents (antibodies and ECL) currently available.

4. The authors should perform confocal and live cell imaging to supplement their findings from Figure 6A.

We apologise for the lack of clarity on this. The data in **Figure 6** is in fact confocal microscopy and live cell imaging. Although we note this in the Methods section under “Confocal Microscopy”, we have now stated this in the results (**page 12, paragraph 2**).

5. The authors demonstrate that Activin A stimulation induces receptor clustering of Type I and Type II receptors to promote Smad 1/5/8 activation but they did not address whether Activin A promotes the overall stability of this complex at the cell surface.

This is an interesting point. However, given the low levels of receptors in the cells, and the complete lack of antibodies, it is not something that we can address at present.

6. Alterations in Smad 1/5 signaling should be supplemented with the expression of Smad 1/5-target genes.

We thank the reviewer for this suggestion and have now quantified the expression of SMAD1/5 target genes *ID1*, *ID3* and *ATOH8* by qPCR in these cells (**new Figure EV1D**). These genes are induced in parental cells, with the induction being more robust and sustained in HET and HOM1 cells.

Referee #2:

In this manuscript the authors explore Activin signaling through ACVR1, also known as ALK2. When ACVR1 was first cloned it was identified as an activin receptor, and then subsequently a BMP receptor. Since then it has become clear that activin can engage ACVR1 to activate Smad1 and Smad5 in certain contexts, as well as ACTR1B/C more generally to induce Smad2/3 activation (see for example, Olson et al., 2020. Biomolecules 2020, 10(4), 519). This dual specificity of activins is dependent on context, since Activins can also antagonize BMP signalling in certain systems. A specific mutation in ACVR1 (R206H) causing fibrodysplasia ossificans progressive (FOP) was shown in 2015 to promote Activin-dependent signaling through Smad1/5. In these models Activin maintains a TGFbeta-like pathway and forms a non-signaling complex with ACVR1 presumably to buffer Smad1

signaling. In contrast, the R206H mutant leads to aberrant activin-dependent activation of Smad1/5. So the prior literature suggests that in certain contexts, particularly with respect to the R206H mutant, Activin A can induce BMP-Smad signaling and disease.

We thank the reviewer for succinctly placing our work in the framework of what is currently known in the field. Before proceeding, we would like to point out that the formation of a non-signalling complex is one possible explanation that has been proposed to account for observations in the field. In fact, we draw the reviewers' attention to the fact in the original paper that suggested these non-signalling complexes (Hatsell et.al. 2015, PMID: 26333933), there was very clear evidence for SMAD1 phosphorylation in ACVR1^{WT} cells (see Figure 1 C, D in that paper) which we will discuss further below. We have now referred to this in the revised text (see **page 15, paragraph 2**).

Here the authors explore ACVR1 signaling downstream of Activin and report that ACVR1 mediates Activin-dependent Smad1/5 activation through a canonical type II receptor-dependent pathway that also employs ACVR1B/C. This is analogous to previous studies showing that in endothelial cells TBR1 mediates recruitment and optimal activation of the ACVR1-like type I receptor, ALK1, to mediate TGFbeta-dependent activation of Smad1/5 in endothelial cells (Goumans et al., Mol. Cell, 2003; EMBO J., 2002). The current manuscript attempts to extend this prior work to an activin pathway, and additionally proposes that as in prior publications, the R206H mutant converts ACVR1 into an activin-dependent BMP receptor. Here the authors add to this work by proposing the mutation might induce receptor clustering and promiscuous kinase activity, which supports previously published structural studies of ACVR1. Overall, this paper explores how the R206H mutation in ACVR1 might function at the molecular level, but very little of the conclusions are all that novel, and rather extend prior observations to this particular model. As noted below, the studies to try and demonstrate an ACVR2:ACVR1B/C:ACVR1 signaling complex in wild type cells are highly problematic. In sum, I'm puzzled by what "new paradigm" is revealed in this manuscript. Furthermore, the paper relies on a single cell line model, which is highly problematic, given the important role of cell context of TGFbeta family dual pathway signaling highlighted above.

The new paradigm referred to in the title is the ability of Activin A to induce pSMAD1/5 via ACVR1^{R206H}, without the requirement for **any other** kinase activity. This is the first time that a single kinase has been shown to be entirely sufficient for TGF- β family signalling in a ligand-dependent manner. This is completely novel and we believe qualifies as a new paradigm of receptor activation in TGF- β family signalling with important consequences for understanding the pathogenesis of diseases involving mutant ACVR1.

In addition, our current paper develops on our 2018 study (Ramachandran et al, PMID 29376829) where we described a new mechanism for TGF- β family receptor activation. There, for the first time we directly demonstrated that type I receptors could activate one another. We showed that activated TGFBR1 could phosphorylate and activate ACVR1 in response to TGF- β . We provided strong evidence that ACVR1 was not in the same heterotetrameric complex with TGFBR1 and TGFBR2. This is in contrast to the conclusions of the Goumans et al paper that the reviewer refers to, which suggested that in endothelial cells TGF- β could activate SMAD1/5 via a hetero-tetrameric complex comprising TGFBR1 (ALK5), ALK1 and TGFBR2.

Our present paper extends our type I receptor transphosphorylation model to Activin. We show in WT cells that Activin activates SMAD1/5 in an analogous way to TGF- β , except that it works via activated ACVR1B/C phosphorylating and activating ACVR1. In the ACVR1^{R206H} cells, this mechanism operates at low doses of Activin A, whilst at high doses of Activin A, ligand-induced ACVR1^{R206H} clustering and auto-activation is the predominant mechanism. We also show that TGF- β can induce auto-activation of ACVR1^{R206H} in a similar way, demonstrating that multiple non-canonical ligands can trigger this mechanism.

We disagree with the reviewer that the model proposed relies on a level of promiscuity. Inherent in promiscuity is an off-target or indiscriminate effect. However, we (PMIDs: 18794361, 29376829) and others (PMID: 11927558, 7806579) have demonstrated that TGF- β -induced pSMAD1/5 is integral to the full biological outcomes mediated by TGF- β and this is likely also the case for Activin A. Indeed, we have now performed qPCR of SMAD1/5 targets genes and found them to be induced in response to Activin A in parental cells and to a greater extent in HET and HOM1 cells (**new Figure EV1D**).

To strengthen our results further, and to answer the criticism that we are relying on a single cell line, we have now additionally used tumour cells from DIPG patients that have either ACVR1^{WT} or ACVR1^{R206H}, obtained from our collaborator, Prof Chris Jones (ICR) to perform key experiments to complement the work currently in the manuscript (**new Figure 3C, new Figure 6I and new Figure EV4B-F**). We show that in cells expressing ACVR1^{R206H}, Activin A-induced pSMAD1/5 is independent of ACVR1B/C as we find in HEK293T cells. Most importantly, we demonstrate Activin A-dependent receptor clustering in these patient cell lines, which is stronger for the line containing ACVR1^{R206H}, and in both cell lines is dependent on the type II receptors ACVR2A/B. Thus, our results from these disease relevant models are in excellent agreement with our data in HEK293T cells.

Specific comments:

1) The biggest issue with this manuscript is that the entire model for wild type signaling is based on inferring a molecular model of receptor interactions by only measuring Smad activation. Critical to support the authors' model is showing a ligand receptor complex comprised of ACVR2:ACVR1B/C:ACVR1, showing that within this complex that ACVR2 phosphorylates ACVR1B/C, and then showing that ACVR1B/C phosphorylates ACVR1. Relying entirely on Smad activation is highly problematic and the results can be interpreted in many different ways. For example, activin could signal through its canonical receptor pathway to modulate an autocrine BMP response. Evidence in the manuscript certainly points to autocrine BMP signaling present in this cell type.

First a technical point. As explained above, analogous to the mechanism we worked out for TGF- β -induced SMAD1/5 phosphorylation, we think that ACVR2A/B, ACVR1B/C and ACVR1 are not in the same tetrameric complex, but that activated ACVR1B/C phosphorylates ACVR1 in the vicinity on the membrane upon ligand-induced receptor clustering.

We thank the reviewer for raising the issue of pathway readouts. After an acute ligand stimulation, pSMAD1/5 **is the most direct** readout of pathway activity as it has the fewest number of steps from extracellular ligand binding to SMAD phosphorylation. At later time points (4–8 h), results can be compounded by the establishment of autocrine/feedback signalling, as pointed out by the reviewer. This is indeed a major issue with previous studies

that have used luciferase reporter assays that need a few hours to allow luciferase levels to accumulate to sufficient levels for detection. However, to complement our pSMAD1/5 kinetics and confirm their biological relevance, we have performed qPCR of SMAD1/5 target genes in these cell lines (**new Figure EV1D**). We find that gene expression mirrors pSMAD1/5 evident by Western blotting, with the induction of *ID1*, *ID3* and *ATOH8* seen in parental cells being more robust and sustained in HET and HOM1 cells.

The reviewer suggests that the activation of pSMAD1/5 by Activin A could be indirect, resulting from canonical Activin receptor signalling establishing autocrine BMP signalling. We think that this is extremely unlikely, given that we can detect induction of pSMAD1/5 in the HOM1 cells within 5' of Activin A treatment (see **Figure 2D** and **Appendix Figure S1A**).

2) In Figure 1 the authors make mutant HEK293 cell lines where they introduce the R206H mutant and KO ACVR1. The results are largely consistent with the known literature. What's puzzling is that while mutant ACVR1 promotes Activin A signaling it does not affect Activin B. At the molecular level it's difficult to reconcile this with the future conclusions about ligand induced ACVR1B/C receptor clustering. Switching to analysis of BMP2 in C and D is puzzling since BMP2 employs additional type I receptors. BMP4:7 should be used for comparative analyses.

The observation that Activin B leads to more robust pSMAD1/5 activation compared to Activin A in ACVR1^{WT} cells was seen on many occasions (over 6 replicates) and we are confident that this is a faithful representation of what happens in HEK293T cells. As a result of this we do not see a dramatic difference in signalling between ACVR1^{WT} and ACVR1^{R206H} for Activin B as we do for Activin A. At present, however, we cannot explain this difference. Mature Activin A and Activin B only share around 74% similarity, leaving open the possibility that mismatches, including in the pre-helix loop and post-helix loop important for receptor binding (Aykul et.al., PMID: 32515349), could result in differences in the ability of these two ligands to bind ACVR1. We do not feel that this observation in any way negates the subsequent body of work on Activin A and ACVR1^{R206H} in this manuscript, as we do not claim that Activin A and Activin B are equivalent. Instead, this observation provides preliminary data for future lines of research on the differences in biology of Activin A and Activin B.

The reviewer raises a very valid point on the use of BMP4/7 as a comparator and we have now replaced **Figure 1D** with a dose response to BMP4/7. Please note that the 20 ng/mL BMP4/7 (and BMP7) response in parental, HET, HOM1 and KO1 cells is presented in **Figure 1A** and we have, therefore, kept the data in **Figure 1C** (20 ng/mL BMP2 in these cell lines) as it provides information on another BMP ligand that signals through a distinct type I receptor.

3) Figure 2 provides a bit more information on Activin A temporal induction of Smad1/5 and adds to the previous studies of Hatsell, Hino and more recently Olsen. The authors continue comparison of activin A with BMP2 in temporal kinetics. This is problematic as BMP2 signaling is not shown to be dependent on ACVR1 in this model. They need to use BMP4:7 dimers, or BMP6 or BMP7, which require ACVR1, particularly as they are using these data to conclude that ACVR1 and the R206H variant do not show substantive differences in trafficking. They then explore the I-SMAD negative feedback loop and propose that Smad2/3-mediated induction of I-SMADs regulate ACVR1. However, BMPs also induce I-SMADs, which was not explored in this model and they did not assess whether I-SMAD KOs

affected extended signaling by the mutant receptor. In particular, the time course of Activin A signaling through mutant receptors in EV3A is extended compared to WT, and so could be consistent with R206H promoting cell surface levels. This highlights a challenge with this part of the manuscript in that it only employs indirect measures of receptor activity to make strong conclusions about WT and R206H having "broadly similar intracellular signaling dynamics and recycling". Directly measuring cell surface and endosomal receptor is required.

We agree with the reviewer that a good control for our Activin A experiments is BMP4/7 heterodimer as it also signals through ACVR1 (as evidenced by the fact that signal induction is lost in the ACVR1 KO cells). We have performed key experiments with this ligand (ligand dose-dependent experiments (**new Figure 1D**); sensitivity to small molecule inhibitors (**new Figures 3B**) and treatment of DIPG cell lines (**new Figure 3C**)). Due to lab access restrictions at our Institute as a result of the COVID pandemic, we have been limited by what we could do practically in the last months so were unable repeat all the experiments with BMP4/7. We do believe, however, that the BMP2 controls that we had performed in the original study are still very valuable as they demonstrate that the presence of ACVR1^{R206H} does not have any indirect effects on other TGF- β family receptors.

In terms of analysing the mechanism of how pSMAD1/5 is terminated, we only did this in the WT cells, as in the HOM1 cells, activation of pSMAD1/5 in the 0–8 h time course is already maximal.

We would really have liked to be able to directly measure receptor levels and receptor trafficking for both ACVR1^{WT} and ACVR1^{R206H}. We thank the reviewer for suggesting this. However, despite over 20 years of looking for antibodies that recognise endogenous ACVR1, we have never found an antibody that either works in Western blots (and thus suitable for biotinylation assays) or in immunostaining. We have investigated whether we can use our fluorescently-tagged Activin to label and track endogenous receptors, but the levels of receptors are so low that the signal-to-noise ratio is insufficient to achieve this, even with a super resolution microscope.

We are able to conclude that the intracellular signalling dynamics are broadly equivalent for ACVR1^{WT} and ACVR1^{R206H} from our Follistatin chase experiment shown in **Figure EV3A**. This is a direct measurement of intracellular signalling. When Follistatin is added 1 h after Activin stimulation, no new receptor binding can occur and thus we are only observing signalling from endosomes. In this case the signalling dynamics in Parental and HOM1 cells are equivalent (**Figure EV3A, lower panel**).

With reference to the cell surface levels of ACVR1^{WT} and ACVR1^{R206H}, we believe that they are present at the cell surface at similar levels, since BMP7 and BMP4/7 (that both signal via ACVR1) activate similar levels of pSMAD1/5 in parental, HET and HOM cells (**Figure 1A and EV1C**). Moreover, we show that after a 2-h stimulation with Activin, followed by a 2-h chase with Follistatin, both parental and HOM1 cells can be readily induced with Activin A, showing that ACVR1^{WT}, ACVR1^{R206H} (and the classic Activin type I receptors ACVR1B and ACVR1C) are still present at the cell surface following an acute Activin A treatment. This tells us that the more transient Activin A response via ACVR1^{WT} is not due to its more rapid depletion from the cell surface compared with ACVR1^{R206H}. Furthermore, after an 8-h Activin stimulation, BMP4/7 can still activate pSMAD1/5 to similar levels in both parental and HOM1

cells (**Figure EV3C**), suggesting that ACVR1 is present at equivalent levels in both cell types at this time point after Activin A stimulation.

4) In Figure 3 the authors present experiments that suggest ACVR1B/C is important for Activin-dependent ACVR1 signaling to BMP Smads. The problem with these experiments is that the cell model displays marginal Activin-dependent Smad1 activation (only about 3 fold at best and about ~10-fold less than the BMP response, see Fig. 1), so the pathway is not active in these cells, which makes interpretation of the results challenging, especially since KO lines are essentially clonal derivatives of the parental. Thus, even slight clonal variation in responsiveness could account for the observations on WT ACVR1 cells in panel B. Of note, recent studies (Aykul et al., eLife, 2020), also in HEK293 cells, show no evidence of activin-dependent activation of Smad1 either by blotting for P-Smad1 or assessing a BMP reporter construct. It's unclear why the current studies show these trace BMP activation events.

It is true that parental cells have a weak pSMAD1/5 response to Activin A compared with HOM cells. However, we feel that this comparison is a major strength of the paper as it demonstrates the significant gain in signalling output conferred by ACVR1^{R206H} at the endogenous locus, as observed in multiple clones. Strikingly, we also show that the Activin A-induced pSMAD1/5 in parental cells is via a distinct mechanism compared to the ACVR1^{R206H} cells.

We refute the assertion that the pSMAD1/5 pathway downstream of Activin A is not active in parental HEK293T cells. In our hands, there is clear pSMAD1/5 in response to Activin A in over 50 experiments in various contexts. In the paper referenced by the reviewer (Aykul et.al. PMID: 32515349), the authors looked at pSMAD1/5 15 minutes after Activin A treatment in HEK293T cells, a time point at which pSMAD1/5 is not evident, in agreement with our observations (**Figure 2D, Appendix Figure S1A**). However, they do observe pSMAD1/5 in response to Activin A in mouse embryonic stem cells after 15 minutes of Activin A treatment (Figure 1 – Figure Supplement 1). Furthermore, in a previous publication from that group, which first demonstrated the sensitivity of ACVR1^{R206H} to Activin A mediated pSMAD1/5, pSMAD1/5 was clearly evident, although unacknowledged, in response to Activin A in ACVR1^{WT} mESCs after 1 h of ligand treatment (Hatsell et.al., PMID: 26333933, Figure 1C, D). Hino et.al., (PMID: 26621707) also observed pSMAD1/5 in ACVR1^{WT} induced mesenchymal stem cells (Figure 1D). Furthermore, in a recent paper in myeloma cell lines, Activin A-dependent pSMAD1/5 was evident (Olson et al., PMID: 32235336).

Thus, we argue that pSMAD1/5 in response to Activin A is a widespread phenomenon in cells that respond to Activin A. Importantly, we now demonstrate using qPCR for target genes that the Activin A-induced pSMAD1/5 we observe in parental cells is fully functional (**new Figure EV1D**). We have now noted these previous instances of pSMAD1/5 in response to Activin A in the discussion (**page 15, paragraph 2**).

Note that panel A is very noisy and certainly shows evidence of SB421542 inhibition of the R206H mutant. In panel B BMP4:7 should be used for the comparison. The data suggesting ACVR1B/C is not required for R206H activity is convincing.

We are pleased that the reviewer finds the data in ACVR1B/C dKO cells convincing.

We thank the reviewer for raising a point that was perhaps not sufficiently clear in the manuscript. The results in **Figure 3** demonstrate that Activin A achieves phosphorylation of SMAD1/5 by two mechanisms depending on dose of Activin A and status of ACVR1. In parental cells, at both doses of Activin A, low or high, Activin A-induced pSMAD1/5 is inhibited by SB-431542, showing that it is entirely dependent on ACVR1B/C. In ACVR1^{R206H} cells treated with a low dose of Activin A, SB-431542 also substantially inhibits induction of pSMAD1/5. Thus, at low dose, Activin A-induced pSMAD1/5 in ACVR1^{R206H} cells is dependent on ACVR1B/C. However, at high doses of Activin A in ACVR1^{R206H} cells, significant pSMAD1/5 persisted even after treatment with SB-431542. This is clear in both the representative westerns and in the quantitations (**Figure 3A, new Figure 3B**). This indicates that at high doses, Activin A *additionally* induces pSMAD1/5 via ACVR1^{R206H} independently of ACVR1B/C. We show later in the paper that the underlying mechanism involves the high dose of Activin A clustering ACVR1^{R206H} to activate it, which is independent of the kinase activity of either ACVR1B/C or the type II receptors. We have now further shown that in the ACVR1^{R206H} DIPG cell line HSJD-DIPG-007, SB-431542 has very little effect on Activin induced pSMAD1/5 although it completely inhibits Activin induced pSMAD2 (**new Figure 3C**) demonstrating that this mechanism also operates in biologically relevant cell lines. We have improved the text in the results (**pages 9 and 10**) in the manuscript to make this clearer.

We agree with the reviewer that BMP4/7 is a better comparator in these experiments, and we have, therefore, performed additional experiments with the small molecule inhibitors with BMP4/7 (**new Figure 3B**). Incidentally, these new experiments contain a repeat of the high dose of Activin A induction to strengthen that data. Furthermore, we used BMP4/7 as the comparator in the DIPG cell lines ICR-B169 and HSJD-DIPG-007 (**new Figure 3C**). As expected, and similar to what we previously observed with BMP2, BMP4/7-induced pSMAD1/5 is inhibited by LDN-193189 and unaffected by SB-431542 in parental, HET and HOM1 cells. We do not feel that BMP4/7 needs to be tested in ACVR1B/C dKO cells as none of the BMPs signal via these receptors.

5). The authors also attempt to show that TGFbeta induces p-Smad1 in MEFs. Unfortunately, the data is even less convincing than the conclusion that Activin A induces P-Smad1. What's the biological significance of such a marginal effect? This experiment is also noisy and the authors should employ BMP4:7 for comparative purposes.

Although weak, we do observe pSMAD1/5 in response to TGF- β in MEFs and this could not be inhibited by SB-431542 in Acvr1^{+R206H} cells. This raises the possibility that TGF- β signalling may also be important in the pathogenesis of ACVR1^{R206H} in FOP and DIPG and represents an avenue for future research. However, to strengthen this work, we have now further shown that in the ACVR1^{R206H} DIPG cell line HSJD-DIPG-007, SB-431542 has very little effect on Activin A-induced pSMAD1/5 although it completely inhibits Activin induced pSMAD2 (**new Figure 3C**) demonstrating that this mechanism also operates in biologically relevant cell lines.

6) Having established that ACVR1B/C are not important for activin-dependent R206H signaling, the authors tackle the role of the upstream type II receptors, ACVR2A/B. They show that type II receptors and their catalytic activity are important for signaling in the wild type scenario, but then show a kinase dead ACVR2A can support R206H signaling. They use optogenetics to induce receptor clusters and show that artificially induced clustering of R206H induces activation. An important observation here that is ignored by the authors is

that in the absence of light the R206H mutant also clusters at higher expression levels. So this mutation provides a propensity for type I receptors to cluster and yield promiscuous autophosphorylation that at endogenous levels is promoted by the type II receptor. This supports previous conclusions about how the mutant converts into an Activin->BMP Smad signaling complex as well as structure-based studies.

The reviewer makes an important point that we did not comment on the light-independent pSMAD1/5 mediated by ACVR1^{R206H} at higher receptor concentrations and we have now done so in the revised manuscript (**page 11, paragraph 2**). However, this does not demonstrate a propensity for ACVR1 (either WT or R206H) to cluster, or to exhibit promiscuous behaviour, but rather highlights the drawbacks of overexpression of receptors that achieves supraphysiological levels of receptors which leads to artificial clustering, a problem that has plagued the field of ACVR1^{R206H} research for many years. Indeed, as a consequence of such overexpression studies, for many years the field considered ACVR1^{R206H} to signal in a ligand-independent fashion, a conclusion we now know is not true. The point we were making with this experiment was that all that is required for ACVR1^{R206H} to activate SMAD1/5 was to be clustered (via light in this case). We go on to show that in the HOM1 cells, this is normally achieved by the ligand-bound type II receptors but is independent of their type II receptor kinase activity.

7). At the top of page 11 the authors make the strange conclusion that since light-induced ACVR1 clusters fail to activate Smad1, then this demonstrates the key requirement for ACVR1B/C. But BMP signaling through ACVR1 does not require ACVR1B/C so there is no reason why ACVR1 in such an artificial system would not activate Smad1 except for the fact no type II kinase is present in these light-induced clusters.

We apologise for the confusion caused by this sentence. What we meant was that because Opto-ACVR1^{WT} was unable to mediate FLAG-SMAD1 phosphorylation, another, upstream kinase was required to activate it. In the case of BMP signalling, as the reviewer points out, this is provided by the type II receptors and in the case of Activin A signalling, we have demonstrated that this is achieved by ACVR1B/C. We therefore sought to confirm this by using Opto-ACVR1B* with Opto-ACVR1^{WT} (**Figure 5C**). We have amended the text in the results (**pages 11 and 12**) to make this clearer.

8) In the final experiments the authors go on to hypothesize that in vivo Activin A induces type II-type I complexes and receptor clustering. I found this phrasing rather odd given that this has been amply demonstrated in the prior literature including structural studies of higher order receptor ligand complexes, all of which is confirmed here, so I don't understand the value of these experiments in the context of the current paper.

While work has previously been performed on TGF- β family receptors, almost all of it has been with overexpressed, tagged receptors as there have been no means available to study endogenous receptor clustering. This paper represents the first time that endogenous TGF- β family receptor clusters have been reported in the literature, a highly significant and very long overdue achievement, especially given the much more advanced state of receptor dynamics elucidated for receptor tyrosine kinases such as EGFR. The work in this manuscript lays the foundation for accelerated study of endogenous receptor dynamics in TGF- β family signalling.

9). Throughout the manuscript there are numerous experiments that lead to important conclusions that appear to have only been done once. Eg Fig. 1A, EV3A et al. I also find the use of mean +/- SEM for experiments done only twice to be inappropriate.

We wish to clarify that none of the experiments in this paper have only been done once; all have been performed 2–3 times (often more), with representative experiments shown where indicated. This has now been noted in the Materials and Methods section, under Quantification and Statistical Analysis. The only figure where we have presented mean +/- SEM of two experiments is **Figure 2A**, where we note there is no significant difference.

Referee #3:

The manuscript by Ramachandran et al. investigates the mechanisms through which the wildtype Activin Type 1 receptor (ACVR1WT) and a "gain-of-function" mutant form (ACVR1R206H) are activated in response to Activin stimulation. Through a series of biochemical experiments, the authors elucidate two distinct mechanisms governing the activation of these receptors. In the case of ACVR1WT, the authors determine that Activin A can bind to ACVR2A/B and ACVR1B/C, which in turn phosphorylates and activates ACVR1WT. A similar mechanism may also activate ACVR1R206H at low activin concentrations. However, at high activin concentrations the authors demonstrate that ACVR1R206H can form direct complexes with ACVR2A/B. In this latter scenario, the catalytic domain of the type II activin receptors (ACVR2A/B) are dispensable, contributing a scaffolding function that enhances signaling via ACVR1R206H.

The ACVR1R206H receptor is implicated in two diseases, which include Fibrodysplasia ossificans progressiva (FOP) and diffuse intrinsic pontine glioma (DIPG). Thus, considerable effort has been invested into determining how the ACVR1R206H receptor is constitutively activated. The current study uses extensive biochemical assays and sophisticated approaches to generate data that supports the novel mechanisms proposed in the paper. This paper will be of interest to researchers in this field. One aspect that is missing from the current study is an attempt to assess if the proposed mechanisms elucidated in this paper are operative in disease models in which this gain of function receptor contributes to the disease (e.g. FOP).

We thank the reviewer for their thoughtful and accurate assessment of our work. To strengthen the mechanistic implications of this study and to answer the criticism that we are relying on a single cell line, we have additionally used tumour cells from DIPG patients that have either ACVR1^{WT} or ACVR1^{R206H}, obtained from our collaborator, Prof Chris Jones (ICR) to perform key experiments to complement the work currently in the manuscript (**new Figure 3C, new Figure 6I and new Figure EV4B-F**). We show that in cells expressing ACVR1^{R206H}, Activin A-induced pSMAD1/5 is independent of ACVR1B/C as we find in HEK293T cells. Most importantly, we demonstrate Activin A-dependent receptor clustering in these patient cell lines, which is stronger for the line containing ACVR1^{R206H}, and in both cell lines is dependent on the type II receptors ACVR2A/B. Thus, our results from these disease relevant models are in excellent agreement with our data in HEK293T cells.

Specific Points:

1) *The authors state that HEK293T cells that are homozygous for the ACVR1R206H mutation exhibit dramatically increased pSmad1/5 levels compared to parental HEK293T*

cells. While this is true for the HOM1 cells in Figure 1, this does not appear to be the case for HOM2 cells (Figure EV1). For example, the pSmad1/5:Actin ratio in parental HEK293T cells is increased 29 fold following Activin A treatment compared to a 253fold increase in HOM1 cells (Figure 1A). In contrast, in Figure EV1C, Activin increases the pSmad1/5:Actin ratio by 13-fold relative to unstimulated HEK293T parental cells and only 10-fold in HOM2 cells. Can the authors comment on this discrepancy?

We thank the Reviewer for noting this. This is a consequence of experimental variability and fold changes are particularly vulnerable to levels of basal signalling. We now have another replicate of the experiment shown in new **Figure EV1C** that shows that HOM2 cells more closely reflect HOM1 cells.

2) In the discussion, the authors speculate that the ACVR1R206H receptor could undergo differential ubiquitination and lysosomal degradation (reduced relative to wildtype ACVR1), which could lead to prolonged signaling. In light of this, it would be important to show that similar levels of ACVR1WT is expressed in wildtype HEK293T cells when compared to ACVR1R206H in HOM1 or HOM2 cells by immunoblot approaches.

We believe that ACVR1^{WT} and ACVR1^{R206H} are present at the cell surface at similar levels, since BMP7 and BMP4/7 (that both signal via ACVR1) achieve similar pSMAD1/5 levels in parental, HET and HOM cells (**Figure 1A**). In addition, after an 8-h Activin stimulation, BMP4/7 can still activate pSMAD1/5 to similar levels in both parental and HOM1 cells (**new Figure EV3C**), suggesting that ACVR1 is present at equivalent levels in both cell types at this time point. We do agree that measuring cell surface and endosomal receptor levels would add great value to this manuscript. However, at present, despite the many years for which the receptors have been known and the vast body of literature on TGF- β family signalling, there are very few antibodies that reproducibly and reliably detect endogenous receptors and there are none that can detect endogenous ACVR1 in HEK293T cells. Therefore, the experiment proposed in this point is not possible at the current time.

3) In line with the second comment, the authors have created a unique dimeric Activin A ligand that is tagged with a fluorophore (His-Activin A-Atto647N: used in Figure 6 and 7). Could live-imaging approaches be used to trace activin-bound ACVR1WT or ACVR1R206H receptors with co-labelling with an early endosomal marker (EEA1) or a lysosomal marker (LAMP1). This would corroborate the conclusions drawn from Figure 3 (no difference between endosomal signaling between the two receptors) and determine if less ACVR1R206H is degraded.

We thank the reviewer for this excellent suggestion, and we have indeed investigated whether we can use our fluorescently-tagged Activin to label and track endogenous receptors. However, the levels of receptors are so low that the signal-to-noise ratio is insufficient to achieve this, even with a super resolution microscope.

4) Non-canonical pathways, beyond Smad1/5, have also been suggested to be activated downstream of ACVR1R206H, most notably p38 activation. Have the authors assessed p-p38 and total p38 levels in parental, HET and HOM cells following Activin stimulation?

Additional pathways activated by ACVR1^{R206H} have not been investigated in this work as we focused on the mechanism of activation of ACVR1^{R206H}. However, the multitude of signalling

pathways initiated by ACVR1^{R206H} is integral to its pathology and warrants further investigation in subsequent studies. It is however, beyond the scope of this current paper.

5) The strength of the paper lies in the rigorous biochemical characterization of these receptor activation mechanisms in a tractable cell model. However, it would strengthen the paper considerably if supporting evidence for this activation model could be obtained using samples in which expression of ACVR1R206H is implicated in the pathology. Indeed, knock in mice harboring the ACVR1R206H allele develop phenotypes reminiscent of those seen in FOP patients. Significant data also indicates that Activin A can contribute to excessive ossification in models of FOP. Could samples (ossified skeletal muscle for example) from the ACVR1R206H knock in model be assessed for complex formation between ACVR2A/B receptors with ACVR1R206H using proximity ligation assays (PLAs). This would depend on antibodies that could detect the type II and type I receptors. One might predict, based on the model, that high PLA signals should be detected between ACVR2A/B and ACVR1R206H. If such samples are impossible to get, could the MEFs be used (described in Figure EV5). PLA assays in +/- ACVR1R206H MEFs would show a low PLA signal between ACVR2A/B and ACVR1R206H under low Activin A conditions but an increased association between the type II receptors and gain of function type I receptor under high activin A concentrations, if the model is correct. A low PLA signal would be expected in the wildtype MEFs. These experiments would add substantially to the paper.

We thank the reviewer for commending the rigour of the experiments performed. To strengthen the mechanistic implications of this study, we have now used tumour cells from DIPG patients that have either ACVR1^{WT} or ACVR1^{R206H} to perform key experiments to complement the data currently in the manuscript (**new Figure 3C, new Figure 6I and new Figure EV4B–F**). We have also performed qPCR to monitor downstream pathway activity (**new Figure EV1D**).

As noted by the reviewer, one major limitation in TGF- β family signalling is the lack of good antibodies that reproducibly and reliably detect endogenous receptors; indeed, there are none that can detect endogenous ACVR1. This unfortunately rules out a PLA approach. Therefore, at present, we cannot study the dynamics of endogenous ACVR1. However, we have used labelled Activin A (His-Activin A-Atto647N) in live cell imaging with Number and Brightness (N&B) analysis to investigate endogenous receptor clustering, and believe this to be a major step forward. We demonstrate that this ligand-induced receptor clustering is dependent on ACVR2A/B in two ways (ACVR2A/B dKO and ligand traps). In the revised manuscript, we have now repeated these experiments with the cells from DIPG patients and importantly, have been able to detect significantly more receptor clustering in the cell line containing ACVR1^{R206H} compared with the line containing ACVR1^{WT} (see **new Figure 6I and new Figure EV4B–F**). Taken together with our data on rescue with ACVR2A/B^{WT} (parental and HOM1 dKO cells) or ACVR2A^{KR} (HOM1 dKO cells) and activation of signalling by optogenetic clustering of ACVR1^{R206H} we believe we have provided substantial data to support our model.

Dear Caroline,

Thank you again for submitting your revised manuscript (EMBOJ-2020-106317R1) to The EMBO Journal, as well as giving us additional input during the preconsultation. As, mentioned, your amended study was sent back to the referees for re-evaluation, and we have received comments from referees #1 and #3, which I enclose below. As you will see, the referees stated that the study has been substantially improved and that major issues have been comprehensively resolved and they are now broadly in favour of publication, pending minor revision. We took note i.p. that referee #1 was now satisfied with your additional response on the ACVR1R206H clustering and cell surface localization aspects. Further, while referee #1's concern on additional RNA-seq analyses is per se well taken, we editorially concluded that these are in our view not required for this study.

Thus, we are pleased to inform you that your manuscript has been accepted in principle for publication in The EMBO Journal.

We do need you to take care of a number of minor issues related to formatting and data representation as detailed below, which should be addressed at re-submission.

Further, I will share additional edits and comments from our production team during the next days to be considered.

Please contact me at any time if you have additional questions related to below points.

Thank you for giving us the chance to consider your manuscript for The EMBO Journal. I look forward to your final revision.

Again, please contact me at any time if you need any help or have further questions.

Kind regards,

Daniel

>> Introduce ORCID IDs for all corresponding authors (A.R.) via our online manuscript system. Please see below for additional information.

>> Recheck callouts and their correct order in the main text for Fig. EV1D, EV2A and EV4B.

>> Please specify author contributions for E.S. .

>> Introduce a data accessibility section stating: 'This study does not include data amenable for deposition in external repositories'.

>> Add the funding information 'Add 'UK Medical Research Council (FC001095) and Wellcome Trust (FC001095)' to our online system.

>> Dataset EV legends need to be removed from the manuscript and zipped with each respective movie file.

>> As we do not offer the option to state, 'data not shown', please amend the data referred to at p.10, or remove the statement.

>> Enter the corresponding author name, journal and manuscript number to the author checklist.

>> Please add the human consent statement on DIPG cell lines also to the Material and Methods cell line part.

Please note that as of January 2016, our new EMBO Press policy asks for corresponding authors to link to their ORCID iDs. You can read about the change under "Authorship Guidelines" in the Guide to Authors here: <http://emboj.embopress.org/authorguide>

In order to link your ORCID iD to your account in our manuscript tracking system, please do the following:

1. Click the 'Modify Profile' link at the bottom of your homepage in our system.
2. On the next page you will see a box half-way down the page titled ORCID*. Below this box is red text reading 'To Register/Link to ORCID, click here'. Please follow that link: you will be taken to ORCID where you can log in to your account (or create an account if you don't have one)
3. You will then be asked to authorise Wiley to access your ORCID information. Once you have approved the linking, you will be brought back to our manuscript system.

We regret that we cannot do this linking on your behalf for security reasons. We also cannot add your ORCID iD number manually to our system because there is no way for us to authenticate this iD number with ORCID.

Thank you very much in advance.

Further information is available in our Guide For Authors:

The revision must be submitted online within 90 days; please click on the link below to submit the revision online before 16th Jun 2021.

Link Not Available

Referee #1:

In this study, Ramachandran et. al investigate the mechanisms by which Activin Type 1 receptor (ACVR1WT) and the mutant (ACVR1R206H) are activated in response to Activin. Through rigorous biochemical and advanced microscopy studies, the authors demonstrated that ACVR1WT and ACVR1R206H activate downstream transcription factors, Smad 1/5 through two distinct mechanisms. ACVR1WT activation is contingent on activin type 1 receptors, ACVR1B/C. ACVR1R206 activation, however, does not required upstream kinases and is activated by Activin A-mediated receptor clustering. Given that the ACVR1R206H mutation has been implicated in two different diseases including Fibrodysplasia ossificans progressive (FOP) and diffuse intrinsic pontine glioma (DIPG), it is of critical importance to understand how ACVRR206H is constitutively activated.

One of the major limitations of the initial submission was a lack of clinical relevance. In the initial submission, the authors state that two different diseases (FOP and DIPG) are driven by a shared ACVR1R206H mutation. However, little to no studies were performed in the context of either disease. For this resubmission, the authors have firmly addressed this concern by demonstrating that Activin A-induced pSmad 1/5 is independent of ACVR1B/C in tumors cells from DIPG patients that have either ACVR1WT or ACVR1R206H mutations. These results support their initials findings in H293T cells and substantially enhance the clinical implications of this study. While the authors' responses placated the majority of my concerns, there are still several points that need to be addressed:

Concerns:

A major limitation of this study was the lack of supporting evidence surrounding ACVR1R206H-mediated enhancement of Smad 1/5/8 activation outside of western blotting. While the authors were able to address this concern by demonstrating that Smad 1/5 target gene induction mirrored the duration and magnitude of Smad 1/5 phosphorylation upon Activin A stimulation in HET and HOM1 cells compared to parental cells, a more conclusive and comprehensive analysis, such as RNA seq, should be used to show that TGFB signaling, specifically Smad 1/5, is the predominant signaling pathway activated upon Activin stimulation in HET and HOM cells. These findings will placate unanimous concerns of whether non-canonical pathways, Smad 1/5 independent pathways, predominate during Activin A stimulation.

Another major limitation of this study was the lack of additional techniques, outside of TIRF microscopy and follastatin chase experiments, to demonstrate ACVR1R206H and ACVR1WT subcellular dynamics. While additional immunofluorescence studies and biotinylation/streptavidin labeling experiments were suggested by myself, Referee #2, and Referee #3, the authors claim that this cannot be performed due to the lack of reliable antibodies that detect endogenous ACVR1. Moreover, the authors suggest that reliance on overexpression systems will alter the physiological balance of receptors and result in artificial clustering and confounding results. Although these are valid and acceptable points, fluorescence imaging demonstrating cell surface localization of ACVR1R206H is critical and will bolster their TIRF findings. A feasible method that could be used to bypass the need for exogenous antibodies is Crispr/Cas9 mediated GFP knock-in at the ACVR1 locus in 293T cells.

Referee #3:

The revised manuscript by Ramachandran et al. (EMBOJ_2020-106317R) has largely addressed the questions I raised during my initial review. The description of cell lines from DIPG patients that express wildtype ACVR1 [ACVR1+/+ (ICR-B169)] or the ACVR1R206H mutant receptor [ACVR1+/R206H (HSJD-DIPG-007)] was a very important addition to the manuscript. The results from these cell models nicely support the data generated from the HEK293T system and extend the conclusions beyond a single cell line into disease relevant models. Additional minor comments were sufficiently addressed as well.

The only experiments that were not addressed by the authors were due to the fact that the necessary reagents are not available (decent antibodies against ACVR1 and ACVR2 receptors) or due to technical limitations of the system (live cell imaging with His-Activin A-Atto647N).

I support publication of the revised manuscript in EMBO J.

Referee #1, additional comment:

The authors made a reasonable argument that their current data do strongly support the conclusion, so I incline to support acceptance of the revised manuscript for publication in the journal.

The authors performed the requested changes.

Dear Caroline,

Thank you for submitting the revised version of your manuscript. I have now evaluated your amended manuscript and concluded that the remaining minor concerns have been sufficiently addressed.

Thus, I am pleased to inform you that your manuscript has been accepted for publication in the EMBO Journal.

Please note that it is EMBO Journal policy for the transcript of the editorial process (containing referee reports and your response letter) to be published as an online supplement to each paper.

Also in case you might NOT want the transparent process file published at all, you will also need to inform us via email immediately. More information is available here:

http://emboj.embopress.org/about#Transparent_Process

Please note that in order to be able to start the production process, our publisher will need and contact you regarding the following forms:

- PAGE CHARGE AUTHORISATION (For Articles and Resources)

[http://onlinelibrary.wiley.com/journal/10.1002/\(ISSN\)1460-2075/homepage/tej_apc.pdf](http://onlinelibrary.wiley.com/journal/10.1002/(ISSN)1460-2075/homepage/tej_apc.pdf)

- LICENCE TO PUBLISH (for non-Open Access)

Your article cannot be published until the publisher has received the appropriate signed license agreement. Once your article has been received by Wiley for production you will receive an email from Wiley's Author Services system, which will ask you to log in and will present them with the appropriate license for completion.

- LICENCE TO PUBLISH for OPEN ACCESS papers

Authors of accepted peer-reviewed original research articles may choose to pay a fee in order for their published article to be made freely accessible to all online immediately upon publication. The EMBO Open fee is fixed at \$5,200 (+ VAT where applicable).

We offer two licenses for Open Access papers, CC-BY and CC-BY-NC-ND.

For more information on these licenses, please visit: <http://creativecommons.org/licenses/by/3.0/> and http://creativecommons.org/licenses/by-nc-nd/3.0/deed.en_US

- PAYMENT FOR OPEN ACCESS papers

You also need to complete our payment system for Open Access articles. Please follow this link and select EMBO Journal from the drop down list and then complete the payment process:

https://authorservices.wiley.com/bauthor/onlineopen_order.asp

Should you be planning a Press Release on your article, please get in contact with embojournal@wiley.com as early as possible, in order to coordinate publication and release dates.

On a different note, I would like to alert you that EMBO Press is currently developing a new format for a video-synopsis of work published with us, which essentially is a short, author-generated film explaining the core findings in hand drawings, and, as we believe, can be very useful to increase visibility of the work. This has proven to offer a nice opportunity for exposure i.p. for the first author(s) of the study. Please see the following link for representative examples:
https://www.embopress.org/video_synopses

If you have any questions, please do not hesitate to call or email the Editorial Office.

Kind regards,

Daniel

Daniel Klimmeck, PhD
Senior Editor
The EMBO Journal
EMBO
Postfach 1022-40
Meyerhofstrasse 1
D-69117 Heidelberg
contact@embojournal.org
Submit at: <http://emboj.msubmit.net>

Corresponding Author Name: Caroline S. Hill and Anassuya Ramachandran

Journal Submitted to: EMBO J

Manuscript Number: EMBOJ-2020-106317R1